# Learning Hierarchical Relational Representations through Relational Convolutions

**Awni Altabaa**                                                         *awni.altabaa@yale.edu*
*Department of Statistics & Data Science*
*Yale University*

**John Lafferty**                                                       *john.lafferty@yale.edu*
*Department of Statistics & Data Science, Wu Tsai Institute*
*Yale University*

**Reviewed on OpenReview:** *https://openreview.net/forum?id=vNZlnznmV2*

## Abstract

An evolving area of research in deep learning is the study of architectures and inductive biases that support the learning of relational feature representations. In this paper, we address the challenge of learning representations of *hierarchical* relations—that is, higher-order relational patterns among groups of objects. We introduce "relational convolutional networks", a neural architecture equipped with computational mechanisms that capture progressively more complex relational features through the composition of simple modules. A key component of this framework is a novel operation that captures relational patterns in groups of objects by convolving graphlet filters—learnable templates of relational patterns—against subsets of the input. Composing relational convolutions gives rise to a deep architecture that learns representations of higher-order, hierarchical relations. We present the motivation and details of the architecture, together with a set of experiments to demonstrate how relational convolutional networks can provide an effective framework for modeling relational tasks that have hierarchical structure.

## 1 Introduction

Objects in the real world rarely exist in isolation; modeling the relationships between them is essential to accurately capturing complex systems. As increasingly powerful machine learning models advance towards building internal "world models," it becomes crucial to explore natural inductive biases to enable efficient learning of relational representations. The computational challenge lies in developing the components required to construct robust, flexible, and progressively more complex relational representations.

An important component of relational cognitive processing in humans is an ability to reason about higher-order relational patterns between groups of objects. To illustrate this, it is instructive to consider experimental tasks from the cognitive psychology literature that probe abstract relational reasoning ability. Consider, for example, the task depicted to the right in Figure 1 which is a variant of a relational "match-to-sample" task (Ferster, 1960; Webb et al., 2021). The subject is presented with a *source* triplet of objects and several *target* triplets, with each triplet having a particular relational pattern. The task is to match the source to a target triplet with the same relational pattern (in this case, the source has an "ABA" pattern that matches the second target). This task requires going beyond reasoning about pairwise relations; the subject must reason about each triplet of objects *as a group*, determine its relational pattern, then compare it to those of the target triplets, inferring the abstract rule in the process. The ability to infer generalizable abstract rules in such tasks is believed to be unique to humans (Fagot et al., 2001).



Figure 1: A variant of a relational match-to-sample task.

Compositionality—used here to mean an ability to compose modules together to build iteratively more complex feature representations—is essential to the success of deep representation learning. For example, in the domain of visual processing, CNNs are able to extract higher-level features (e.g., textures and object-specific features) by composing simpler feature maps (Zeiler and Fergus, 2014), resulting in a flexible architecture for computing "features of features". In contrast, existing work on relational representation learning has primarily been limited to "shallow" first-order architectures (e.g., only explicitly capturing pairwise relations).

In this work, we propose **_relational convolutional networks_** as a compositional framework for learning hierarchical relational representations. The key to the framework involves formalizing the concept of convolving learnable templates of a relational pattern against a larger relation tensor. This operation produces a sequence of vectors representing the relational pattern within each group of objects. Crucially, composing relational convolutions captures higher-order relational features—i.e., relations between relations. Specifically, our proposed architecture introduces the following concepts and computational mechanisms.

- **_Graphlet filters._** A "graphlet filter" is a template for the pattern of relations between a (small) collection of objects. Since pairwise relations can be viewed as edges on a graph, the term "graphlet" is used to refer to a subgraph, and the term "filter" is used to refer to a learnable template or pattern.

- **_Relational convolutions._** We formalize a notion of _relational_ convolution, analogous to spatial convolutions in CNNs, where a graphlet filter is matched against the relations within _groups_ of objects to obtain a representation of the relational pattern in different groupings of the input.

- **_Grouping mechanisms._** For large problem instances, considering relational convolutions across all object combinations would be intractable. To achieve scalability, we introduce a learnable grouping mechanism based on attention which identifies the relevant groups that should be considered for the downstream task.

- **_Compositional relational modules._** The proposed architecture supports composable modules, where each module has learnable graphlet filters and groups. This enables learning higher-order relationships between objects—relations between relations.

The components of the architecture are presented in detail in Sections 2 and 3, and a schematic of the proposed architecture is shown in Figure 2. In a series of experiments, we show how relational convolutional networks provide a powerful framework for relational learning. We first carry out experiments on the "relational games" benchmark for relational reasoning proposed by Shanahan et al. (2020), which consists of a suite of binary classification tasks for identifying abstract relational rules between a set of geometric objects represented as images. We next carry out experiments on a version of the _Set_ card game, which requires processing of higher-order relations across multiple attributes. We find that relational convolutional networks outperform Transformers, graph neural networks, and existing relational architectures. These results demonstrate that both compositionality and relational inductive biases are essential for efficiently learning representations of complex higher-order relations.

## 1.1 Related Work

To place our framework in the context of previous work, we briefly discuss related forms of relational learning below, pointing first to the review of relational learning inductive biases by Battaglia et al. (2018).

Graph neural networks (GNNs) are a class of neural network architectures which operate on graphs and process "relational" data (e.g., Niepert et al., 2016; Kipf and Welling, 2017; Schlichtkrull et al., 2018; Veličković et al., 2018; Kipf et al., 2018; Xu et al., 2019). A defining feature of GNN models is their use of a form of neural message-passing, wherein the hidden representation of a node is updated as a function of the hidden representations of its neighbors on a graph (Gilmer et al., 2017). Typical examples of tasks that GNNs are applied to include node classification, graph classification, and link prediction (Hamilton, 2020).

In GNNs, the 'relations' are given to the model as input via edges in a graph. In contrast, our architecture, as well as the relational architectures described below, operate on collections of objects without any relations given as input. Instead, such relational architectures must infer the relevant relations from the objects themselves. Still, graph neural networks can be applied to these relational tasks by passing in the collection of objects along with a complete graph.

Several works have proposed architectures with the ability to model relations by incorporating an attention mechanism (e.g., Vaswani et al., 2017; Veličković et al., 2018; Santoro et al., 2018; Zambaldi et al., 2018; Locatello et al., 2020). Attention mechanisms, such as self-attention in Transformers (Vaswani et al., 2017), model relations between objects implicitly as an intermediate step in an information-retrieval operation to update the representation of each object as a function of its context.

There also exists a growing literature on neural architectures that aim to explicitly model relational information between objects. An early example is the relation network proposed by Santoro et al. (2017), which produces an embedding representation for a set of objects based on aggregated pairwise relations. Shanahan et al. (2020) proposes the PrediNet architecture, which aims to learn relational representations that are compatible with predicate logic. Kerg et al. (2022) proposes CoRelNet, a simple architecture based on 'similarity scores' that aims to distill the relational inductive biases discovered in previous work into a minimal architecture. Altabaa et al. (2024) and Altabaa and Lafferty (2024b) explore relational inductive biases in the context of Transformers, and propose a view of relational inductive biases as a type of selective "information bottleneck" which disentangles relational information from object-level features. Webb et al. (2024) provides a cognitive science perspective on this idea, arguing that a relational information bottleneck may be a mechanism for abstraction in the human mind.

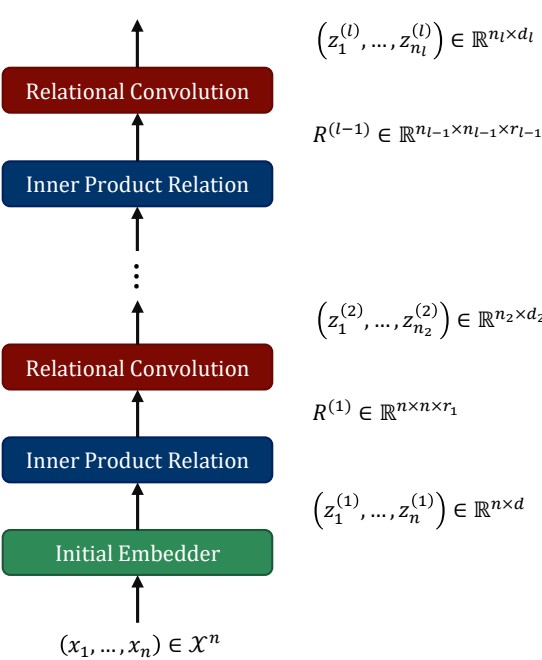

Figure 2: Proposed architecture for relational convolutional networks. Hierarchical relations are modeled by iteratively computing pairwise relations between objects and convolving the resultant relation tensor with graphlet filters representing templates of relations between groups of objects.

## 2    Multi-Dimensional Inner Product Relation Module

A relation function maps a pair of objects $x, y \in \mathcal{X}$ to a vector that represents the relationship between them. For example, a relation may represent comparisons along different attributes of the two objects, such as "$x$ has the same color as $y$, $x$ is larger than $y$, and $x$ is to the left of $y$". In principle, this can be modeled by an arbitrary learnable function on the concatenation of the two objects' feature representations. For example, Santoro et al. (2017) use multilayer perceptrons (MLPs) to model relations by processing the concatenated feature vectors of object pairs. However, this approach lacks crucial inductive biases. While it is theoretically capable of modeling relations, it imposes no constraints to ensure that the learned pairwise function reflects meaningful relational patterns. In particular, it entangles the feature representations of the two objects without explicitly comparing their features.

Following previous work (e.g., Vaswani et al., 2017; Webb et al., 2021; Kerg et al., 2022; Altabaa et al., 2024), we propose modeling pairwise relations between objects via *inner products* of feature maps. This introduces added structure to the pairwise function that explicitly incorporates a comparison operation (the inner product). The advantage of this approach is that it provides added pressure to learn explicitly relational representations, disentangling relational information from attributes of individual objects, and inducing a geometry on the object space $\mathcal{X}$. For example, in the symmetric case, the inner product relation $r(x, y) = \langle \phi(x), \phi(y) \rangle$ satisfies symmetry, positive definiteness, and induces a pseudometric on $\mathcal{X}$. The triangle inequality of the pseudometric expresses a transitivity property—if $x$ is related to $y$ and $y$ is related to $z$, then $x$ must be related to $z$.

More generally, we can allow for multi-dimensional relations by having multiple encoding functions, each extracting a feature to compute a relation on. Furthermore, we can allow for asymmetric relations by having

different encoding functions for each object. Hence, we model relations by

$$r(x, y) = \left( \langle \phi_1(x), \psi_1(y) \rangle, \ldots, \langle \phi_{d_r}(x), \psi_{d_r}(y) \rangle \right), \tag{1}$$

where $\phi_1, \psi_1, \ldots, \phi_{d_r}, \psi_{d_r}$ are learnable functions. The intuition is that, for each dimension, the encoders extract, or 'filter' out, particular attributes of the objects and the inner products compute similarity across each attribute. A relation, in this sense, is similarity across a particular attribute. In the asymmetric case, the attributes extracted from the two objects are different, resulting in an asymmetric relation where one attribute of the first object is compared with a different attribute of the second object. For example, this can model relations of the form "$x$ is brighter than $y$" (an antisymmetric relation).

Altabaa and Lafferty (2024a) analyzes the function approximation properties of neural relation functions of the form of Equation (1). In particular, the function class of inner products of neural networks is characterized in both the symmetric case and the asymmetric case. In the symmetric case (i.e., $\phi = \psi$), it is shown that inner products of MLPs are universal approximators for symmetric positive definite kernels. In the asymmetric case, inner products of MLPs are universal approximators for continuous bivariate functions. The efficiency of approximation is characterized in terms of a bound on the number of neurons needed to achieve a particular approximation error.

To promote weight sharing, we can have one common non-linear map $\phi$ shared across all dimensions together with different linear projections for each dimension of the relation. That is, $r : \mathcal{X} \times \mathcal{X} \to \mathbb{R}^{d_r}$ is given by

$$r(x, y) = \left( \langle W_1^k \phi(x), W_2^k \phi(y) \rangle \right)_{k \in [d_r]}, \tag{2}$$

where the learnable parameters are $\phi$ and $W_1^k, W_2^k, k \in [d_r]$. The non-linear map $\phi : \mathcal{X} \to \mathbb{R}^{d_\phi}$ may be an MLP, for example, and $W_1^k, W_2^k$ are $d_{\text{proj}} \times d_\phi$ matrices. The class of functions realizable by Equation (2) is the same as Equation (1) but enables greater weight sharing.

The "Multi-dimensional Inner Product Relation" (MD-IPR) module receives a sequence of objects $(x_1, \ldots, x_n)$ as input and models the pairwise relations between them by Equation (2), returning an $n \times n \times d_r$ relation tensor, $R[i, j] = r(x_i, x_j)$, describing the relations between each pair of objects.

## 3 Relational Convolutions with Graphlet Filters

### 3.1 Relational Convolutions with Discrete Groups

In this section, we formalize a *relational convolution* operation which processes pairwise relations between objects to produce representations of the relational patterns within groups of objects. Suppose that we have a sequence of objects $(x_1, \ldots, x_n)$ and a relation tensor $R$ describing the pairwise relations between them, obtained by an MD-IPR layer via $R[i, j] = r(x_i, x_j)$. The key idea is to learn a *template* of relations between a small set of objects, and to "convolve" the template with the relation tensor, matching it against the relational patterns in different groups of objects. This transforms the relation tensor into a sequence of vectors, each summarizing the relational pattern in some group of objects. Crucially, this can now be composed with another relational layer to compute *higher-order* relations—i.e., relations on relations.

Fix some filter size $s < n$, where $s$ is a hyperparameter of the relational convolution layer. One 'filter' of size $s$ is given by the *graphlet filter* $f_1 \in \mathbb{R}^{s \times s \times d_r}$. This is a template for the pairwise relations between a group of $s$ objects. Since pairwise relations can be viewed as edges on a graph, we use the term "graphlet filter" to refer to a template of pairwise relations between a small set of objects. Let $g \subset [n]$ be a group of $s$ objects among $(x_1, \ldots, x_n)$. Then, denote the relation sub-tensor associated with this group by $R[g] := [R[i, j]]_{i,j \in g}$. We define the 'relational inner product' between this relation subtensor and the filter $f_1$ by

$$\langle R[g], f_1 \rangle_{\text{rel}} := \sum_{i,j \in g} \sum_{k \in [d_r]} R[i, j, k] f_1[i, j, k]. \tag{3}$$

This is simply the standard inner product in the corresponding Euclidean space $\mathbb{R}^{s^2 d_r}$. This quantity represents how much the relational pattern in $g$ matches the template $f_1$.

In a relational convolution layer, we learn $n_f$ different filters. Denote the collection of filters by $\boldsymbol{f} = (f_1, \ldots, f_{n_f}) \in \mathbb{R}^{s \times s \times d_r \times n_f}$, which we call a *graphlet filter*. We define the relational inner product of a relation

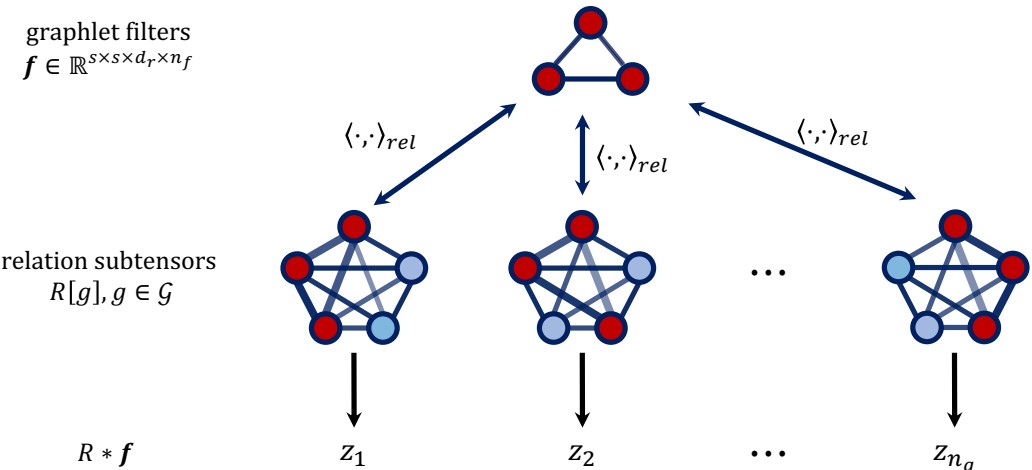

Figure 3: A depiction of the relational convolution operation. A graphlet filter $\boldsymbol{f}$ is compared to the relation subtensor in each group of objects, producing a sequence of vectors summarizing the relational pattern within each group. The groups can be differentiably learned through an attention mechanism.

subtensor $R[g]$ with the graphlet filters $\boldsymbol{f}$ as the $n_f$-dimensional vector consisting of the relational inner products with each individual filter,

$$\langle R[g], \boldsymbol{f} \rangle_{\mathrm{rel}} := \begin{pmatrix} \langle R[g], f_1 \rangle_{\mathrm{rel}} \\ \vdots \\ \langle R[g], f_{n_f} \rangle_{\mathrm{rel}} \end{pmatrix} \in \mathbb{R}^{n_f}. \tag{4}$$

This vector summarizes various aspects of the relational pattern within a group, captured by several different filters[1]. Each filter corresponds to one dimension. This is reminiscent of convolutional neural networks, where each filter gives us one channel in the output tensor.

For a given group $g \subset [n]$, the relational inner product with a graphlet filter, $\langle R[g], \boldsymbol{f} \rangle_{\mathrm{rel}}$, gives us a vector summarizing the relational patterns inside that group. Let $\mathcal{G}$ be a set of groupings of the $n$ objects, each of size $s$. The relational convolution between a relation tensor $R$ and a relational graphlet filter $\boldsymbol{f}$ is defined as the sequence of relational inner products with each group in $\mathcal{G}$

$$R * \boldsymbol{f} := \left( \langle R[g], \boldsymbol{f} \rangle_{\mathrm{rel}} \right)_{g \in \mathcal{G}} \equiv \left( z_1, \dots, z_{|\mathcal{G}|} \right) \in \mathbb{R}^{|\mathcal{G}| \times n_f} \tag{5}$$

In this section, we assumed that $\mathcal{G}$ was given. If some prior information is known about what groupings are relevant, this can be encoded in $\mathcal{G}$. Otherwise, if $n$ is small, $\mathcal{G}$ can be all possible combinations of size $s$. However, when $n$ is large, considering all combinations will be intractable. In the next subsection, we consider the problem of *differentiably learning* the relevant groups.

## 3.2 Relational Convolutions with Group Attention

In the above formulation, the groups are 'discrete'. Having discrete groups can be desirable for interpretability if the relevant groupings are known a priori or if considering every possible grouping is computationally and statistically feasible. However, if the relevant groupings are not known, then considering all possible combinations results in a rapid growth of the number of objects at each layer.

In order to address these issues, we can explicitly model and *learn* the relevant groups. This allows us to control the number of objects in the output sequence of a relational convolution such that only relevant groups are considered. We propose modeling groups via an *attention* operation.

---

[1]We have overloaded the notation $\langle \cdot, \cdot \rangle_{\mathrm{rel}}$, but will use the convention that a collection of filters is denoted by a bold symbol (e.g., $\boldsymbol{f}$ vs $f_i$) to distinguish between the two forms of the relational inner product.

Consider the input $(x_1, \ldots, x_n)$, $x_i \in \mathbb{R}^d$. Let $n_g$ be the number of groups to be learned and $s$ be the size of the graphlet filter (and hence the size of each group). These are hyperparameters of the model that we control. For each group $g \in [n_g]$, we learn $s$ different *queries*, $\{q_i^g\}_{g \in [n_g], i \in [s]}$, that will be used to retrieve a group of size $s$ via attention. The $i$-th object in the $k$-th group is retrieved as follows,

$$
\begin{aligned}
\bar{x}_i^g &= \sum_{j=1}^n \alpha_{ij}^g x_j, & g \in [n_g], i \in [s], \\
\alpha_{ij}^g &= \frac{\exp\left(\beta \left\langle q_i^g, \texttt{key}(x_j) \right\rangle\right)}{\sum_{k=1}^n \exp\left(\beta \left\langle q_i^g, \texttt{key}(x_k) \right\rangle\right)}, & g \in [n_g], i \in [s], j \in [n]
\end{aligned}
\tag{6}
$$

where $\bar{x}_i^g$ is the $i$-th object retrieved in the $g$-th group, $q_i^g$ is the query for retrieving the $i$-th object in the $g$-th group, $\texttt{key}(x_j)$ is the key associated with the object $x_j$, and $\beta$ is a temperature scaling parameter.

The $\texttt{key}$ for each object is computed as a function of its position, features, and/or context. For example, to group objects based on their position, the key can be a positional embedding, $\texttt{key}(x_i) = PE_i$. To group based on features, the $\texttt{key}$ can be a linear projection of the object's feature vector, $\texttt{key}(x_i) = W_k x_i$. To group based on both position and features, the $\texttt{key}$ can be a sum or concatenation of the above. Finally, computing keys after a self-attention operation allows objects to be grouped based on the context in which they occur.

The relation subtensor $\bar{R}[g] \in \mathbb{R}^{s \times s \times d_r}$ for each group $g \in [n_g]$ is computed using a shared MD-IPR layer $r(\cdot, \cdot)$,

$$
\bar{R}[g] = \left[ r(\bar{x}_i^g, \bar{x}_j^g) \right]_{i,j \in [s]}.
\tag{7}
$$

The relational convolution is computed as before via,

$$
\bar{R} * \boldsymbol{f} \equiv \left( \left\langle \bar{R}[g], \boldsymbol{f} \right\rangle_{\text{rel}} \right)_{g \in [n_g]}.
\tag{8}
$$

Overall, relational convolution with group attention can be summarized as follows: 1) learn $n_g$ groupings of objects, retrieving $s$ objects per group; 2) compute the relation tensor of each group using an MD-IPR module; 3) compute a relational convolution with a learned set of graphlet filters $\boldsymbol{f}$, producing a sequence of $n_g$ vectors each describing the relational pattern within a (learned) group of objects.

**Computing input-dependent queries.** In the simplest case, the query vectors are simply learned parameters of the model, representing a fixed criterion for selecting the $n_g$ groups. The queries can also be produced in an input-dependent manner. There are many ways to do this. For example, the input $(x_1, \ldots, x_n)$ can be processed with some sequence or set embedder (e.g., through a self-attention operation) producing a vector embedding that can be mapped to different queries $\{q_i^g\}_{i,g}$ using learned linear maps.

**Entropy regularization.** Intuitively, we would ideally like the group attention scores in Equation (6) to be close to discrete assignments. To encourage the model to learn more structured group assignments, we add an entropy regularization to the loss function, $\mathcal{L}_{\texttt{entr}} = (n_g \cdot s)^{-1} \sum_{g,i} H(\alpha_{i,\cdot}^g)$, where $H(\alpha_{i,\cdot}^g) = -\sum_j \alpha_{ij}^g \log(\alpha_{ij}^g)$ is the Shannon entropy. As a heuristic, this regularization can be scaled by a factor proportional to $\log(\texttt{n\_classes})/\log(n)$ so that it doesn't dominate the underlying task's loss. Sparsity regularization in neural attention has been explored in several previous works, including through entropy regularization (e.g., Niculae and Blondel, 2017; Martins et al., 2020; Attanasio et al., 2022).

**Symmetric relational inner products.** So far, we considered *ordered* groups. That is, the relational pattern computed by the relational inner product $\langle R[g], \boldsymbol{f} \rangle_{\text{rel}}$ for the group $(1, 2, 3)$ is different from the group $(2, 3, 1)$. In some scenarios, symmetry in the representation of the relational pattern is a useful inductive bias. To capture this, we define a symmetric variant of the relational inner product that is invariant to the ordering of the elements in the group. This can be done by pooling over all permutations in the group. In particular, we suggest max-pooling or average-pooling, although any set-aggregator would be valid. We define the permutation-invariant relational inner product as

$$
\langle R[g], \boldsymbol{f} \rangle_{\text{rel,sym}} := \text{Pool}\left( \left\{ \langle R[g'], \boldsymbol{f} \rangle_{\text{rel}} : g' \in g! \right\} \right),
\tag{9}
$$

where $g!$ denotes the set of permutations of the group $g$, and pooling is done independently across dimensions.

**Computational efficiency.** Equation ([6](#)) can be computed in parallel with $\mathcal{O}(n \cdot n_g \cdot s \cdot d)$ operations. When the hyperparameters of the model are fixed, this is linear in the sequence length $n$. Equation ([7](#)) can be computed in parallel via efficient matrix multiplication with $\mathcal{O}(n_g \cdot s^2 \cdot d_r \cdot d_{\text{proj}})$ operations. Finally, Equation ([8](#)) can be computed in parallel with $\mathcal{O}(n_g \cdot s^2 \cdot d_r \cdot n_f)$ operations. The latter two computations do not scale with the number of objects in the input, and are only a function of the hyperparameters of the model.

### 3.3 Deep Relational Architectures by Composing Relational Convolutions

A relational convolution block (including a MD-IPR module) is a simple neural module that can be composed to build a deep architecture for learning iteratively more complex relational feature representations.

Following the notation in Figure [2](#), let $n_\ell$ denote the number of objects and $d_\ell$ the object dimension at layer $\ell$. A relational convolution block receives as input a sequence of objects of shape $n_\ell \times d_\ell$ and returns a sequence of objects of shape $n_{\ell+1} \times d_{\ell+1}$ representing the relational patterns among groupings of objects. The output dimension $d_{\ell+1}$ corresponds to the number of graphlet filters $n_f$, and is a hyperparameter. The sequence length $n_{\ell+1}$ corresponds to the number of groups, and is $|\mathcal{G}|$ in the case of given discrete groups (Section [3.1](#)) or a hyperparameter $n_g$ in the case of learned groups via group attention (Section [3.2](#)). Each composition of a relational convolution block computes relational features of one degree higher (i.e., relations between relations).

A common recipe for building modern deep learning architectures is by using residual connections (He et al., [2016](#)) and normalization (Ba et al., [2016](#)). This can be achieved for relational convolutional networks by fixing the number of groups $n_g$ and number of filters $n_f$ hyperparameters to be the same across all layers, such that the input shape and output shape remain the same. Then, letting $H_\ell$ denote the hidden representation at layer $\ell$, the overall architecture becomes $H_{\ell+1} = \text{Norm}(H_\ell + W_{\ell+1}\text{RelConvBlock}(H_\ell))$, where $W_{\ell+1}$ is a linear transformation that controls where information is written to in the residual stream. This ResNet-style architecture allows for the hidden representation to encode relational information at multiple layers of hierarchy, retaining the information at shallower layers. Additionally, we can insert MLP layers to process the relational representations before the next relational convolution layer. In this paper, we limit our exploration to relatively shallow networks of the form $H_{\ell+1} = \text{RelConvBlock}(H_\ell)$.

## 4 Experiments

In this section, we empirically evaluate the proposed *relational convolutional network* architecture (abbreviated RelConvNet) to assess its effectiveness at learning relational tasks. We compare this architecture to several existing relational architectures as well as general-purpose sequence models. The common input to all models is a sequence of objects $X = (x_1, \ldots, x_n) \in \mathbb{R}^{n \times d}$. We evaluate against the following baselines.

- ***Transformer*** (Vaswani et al., [2017](#)). The Transformer is a powerful general-purpose sequence model. It consists of alternating self-attention and multi-layer perceptron blocks. Self-attention performs an information retrieval operation, which updates the internal representation of each object as a function of its context. Dot product attention is computed via $X' \leftarrow \text{Softmax}((XW_q)(XW_k)^\intercal)W_v X$, and the MLP is applied independently on each object's internal representation. The attention scores computed as an intermediate step in dot-product attention can perhaps be thought of as relations that determine what information to retrieve.

- ***PrediNet*** (Shanahan et al., [2020](#)). The PrediNet architecture is an explicitly relational architecture inspired by predicate logic. At a high-level, the PrediNet architecture computes $j$ relations between $k$ pairs of objects. The $k$ pairs of objects are selected via a learned attention operation. The "$j$ relations" refer to a difference between $j$-dimensional embeddings of the selected objects. More precisely, for each head $h \in [k]$, a pair of objects $E_1^h, E_2^h \in \mathbb{R}^d$ is retrieved via an attention operation, and the final output of PrediNet is a set of difference relations given by $D^h = E_1^h W_s - E_2^h W_s$.

- ***CoRelNet*** (Kerg et al., [2022](#)). The CoRelNet architecture is proposed as a minimal relational architecture distilling the core inductive biases that the authors argue are important for relational tasks. The CoRelNet module simply computes inner products between object representations and applies Softmax normalization, returning an $n \times n$ "similarity matrix". That is, the objects $X = (x_1, \ldots, x_n)$ are processed independently to produce embeddings $Z = (z_1, \ldots, z_n)$, and the

similarity matrix is computed as $R = \text{Softmax}(ZZ^\intercal)$. The similarity matrix $R$ is then flattened and passed through an MLP to produce the final output.

- **_Graph Neural Networks_**. Graph neural networks are a class of neural network architectures which operate on graphs-structured data. A graph neural network typically receives two inputs: a graph described by a set of edges, and feature vectors for each node in the graph. GNNs can be described through the unifying framework of neural message-passing. Under this framework, graph-structured data is processed through an iterative message-passing operation given by $h_i^{(l+1)} \leftarrow \text{Update}(h_i^{(l)}, \{h_j^{(l)}\}_{j \in \mathcal{N}(i)})$, where $h_i^{(0)} \leftarrow x_i$. That is, each node's internal representation is iteratively updated as a function of its neighborhood. Here, Update is parameterized by a neural network, and the variation between different GNN architectures lies in the architectural design of this update process. We use Graph Convolution Networks (Kipf and Welling, 2017), Graph Attention Networks (Veličković et al., 2018), and Graph Isomorphism Networks (Xu et al., 2019) as representative GNN baselines.

- **_CNN_**. As a non-relational baseline, we test a regular convolutional neural network which processes the raw image input. The central modules in the baselines above receive an object-centric representation as input. That is, a sequence of vector embeddings produced by a small CNN each corresponding to one of the $n$ objects in the input. Here, instead, a deeper CNN model processes the raw image input representing the entire "scene" in an end-to-end manner.

## 4.1 Relational Games

The _relational games_ dataset was contributed as a benchmark for relational reasoning by Shanahan et al. (2020). It consists of a family of binary classification tasks for identifying abstract relational rules between a set of objects represented as images. The object images depict simple geometric shapes and consist of three different splits with different visual styles for evaluating out-of-distribution generalization, referred to as "pentominoes", "hexominoes", and "stripes". The input is a sequence of objects arranged in a $3 \times 3$ grid. Each task corresponds to some relationship between the objects, and the target is to classify whether the relationship holds among the objects in the input or not (see Figure 4).

In our experiments, we evaluate out-of-distribution generalization by training on the pentominoes objects and evaluating on the hexominoes and stripes objects. The input to the models is presented as a sequence of 9 objects, with each object represented as a $12 \times 12 \times 3$ RGB image. All models share the common architecture $(x_1, \ldots, x_n) \rightarrow \texttt{CNN} \rightarrow \{\cdot\} \rightarrow \texttt{MLP} \rightarrow \hat{y}$, where $\{\cdot\}$ indicates the central module being tested. In all models, the objects are first processed independently by a CNN with a shared architecture. The processed objects are then passed to the central module of the model. The final prediction is produced by an MLP with a shared architecture. In this section, we focus our comparison on four models: RelConvNet (ours), CoRelNet (Kerg et al., 2022), PrediNet (Shanahan et al., 2020), and a Transformer (Vaswani et al., 2017)[2]. The pentominoes split is used for training, and the hexominoes and stripes splits are used to test out-of-distribution generalization after training. We train for 50 epochs using the categorical cross-entropy loss and the Adam optimizer with learning rate 0.001, $\beta_1 = 0.9, \beta_2 = 0.999, \epsilon = 10^{-7}$, and batch size 512. For each model and task, we run 5 trials with different random seeds. Appendix A describes further experimental details about the architectures and training setup.

**Out-of-distribution generalization.** Figure 5 reports model performance on the two hold-out object sets after training. On the hexominoes objects, which are similar-looking to the pentominoes objects used for training, RelConvNet and CoRelNet do nearly perfectly. PrediNet and the Transformer do well on the simpler tasks, but struggle with the more difficult 'match pattern' task. The 'stripes' objects are visually more distinct from the objects in the training split, making generalization more difficult. We observe an overall drop in performance for all models. The drop is particularly dramatic for CoRelNet[3]. The separation between RelConvNet and the other models is largest on the 'match pattern' task of the stripes split (the most difficult task and the most difficult generalization split). Here, RelConvNet maintains a mean accuracy

---

[2]The GNN baselines failed to learn the relational games tasks in a way that generalizes, often severely overfitting. For clarity of presentation, we defer results on the GNN baselines to Appendix A.

[3]The experiments in Kerg et al. (2022) on the relational games benchmark use a technique called "context normalization" (Webb et al., 2020) as a preprocessing step. We choose not to use this technique since it is an added confounder. We discuss this choice in Appendix C.

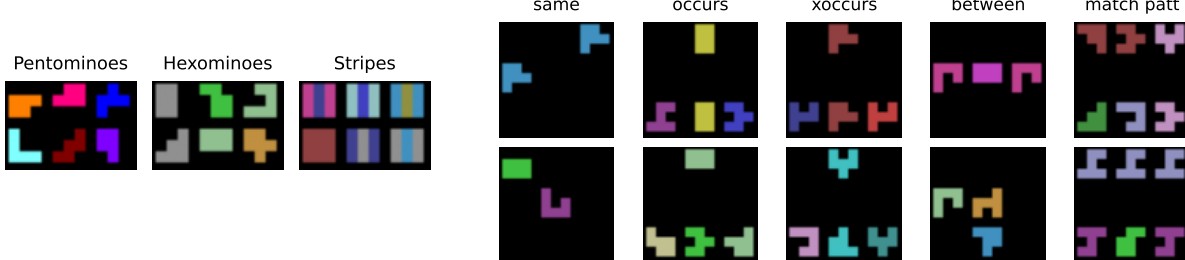

Figure 4: Relational games dataset. **Left** Examples of objects from each split. **Right** Examples of problem instances for each task. The first row is an example where the relation holds and the second row is an example where the relation does not hold.

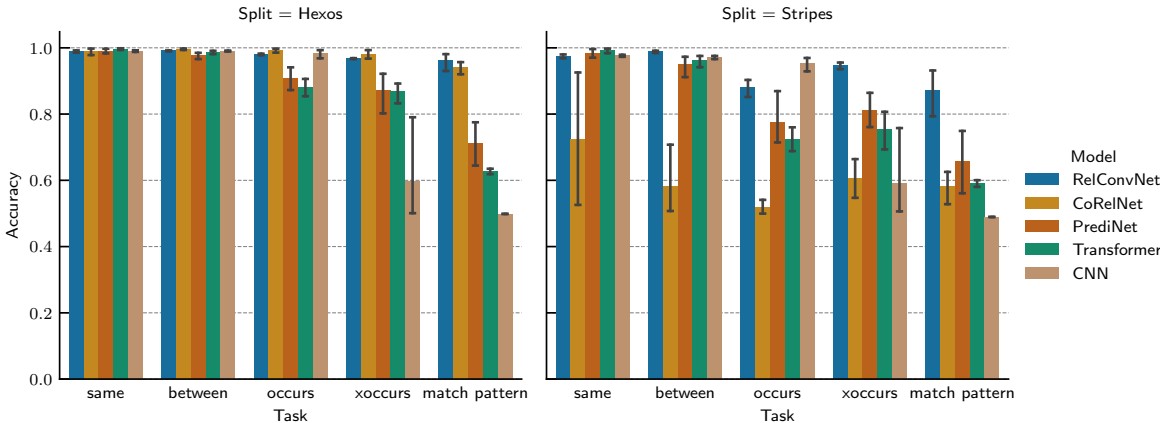

Figure 5: Out-of-distribution generalization on hold-out object sets. Bar heights indicate the mean over 5 trials and the error bars indicate a bootstrap 95% confidence interval.

of 87% while the other models drop below 65%. We attribute this to RelConvNet's ability to naturally represent higher-order relations and model groupings of objects. The CNN baseline learns the easier 'same', 'between', and 'occurs' tasks nearly perfectly, but completely fails to learn the more difficult 'xoccurs' and 'match pattern' tasks. This hard boundary suggests that explicit relational architectural inductive biases are necessary for learning more difficult relational tasks.

**Data efficiency.** We observe that the relational inductive biases of RelConvNet, and relational models more generally, grant a significant advantage in sample-efficiency. Figure 6 shows the training accuracy over the first 2,000 batches for each model. RelConvNet, CoRelNet, and PrediNet are explicitly relational architectures, whereas the Transformer is not. The Transformer is able to process relational information through its attention mechanism, but this information is entangled with the features of individual objects (which, for these relational tasks, is extraneous information). The Transformer consistently requires the largest amount of data to learn the relational games tasks. PrediNet tends to be more sample-efficient. RelConvNet and CoRelNet are the most sample-efficient, with RelConvNet only slightly more sample-efficient on most tasks.

On the 'match pattern' task, which is the most difficult, RelConvNet is significantly more sample-efficient. We attribute this to the fact that RelConvNet is able to model higher-order relations through its relational convolution module. The 'match pattern' task can be thought of as a second-order relational task—it involves computing the relational pattern in each of two groups, and comparing the two relational patterns. The relational convolution module naturally models this kind of situation since it learns representations of the relational patterns within groups of objects.

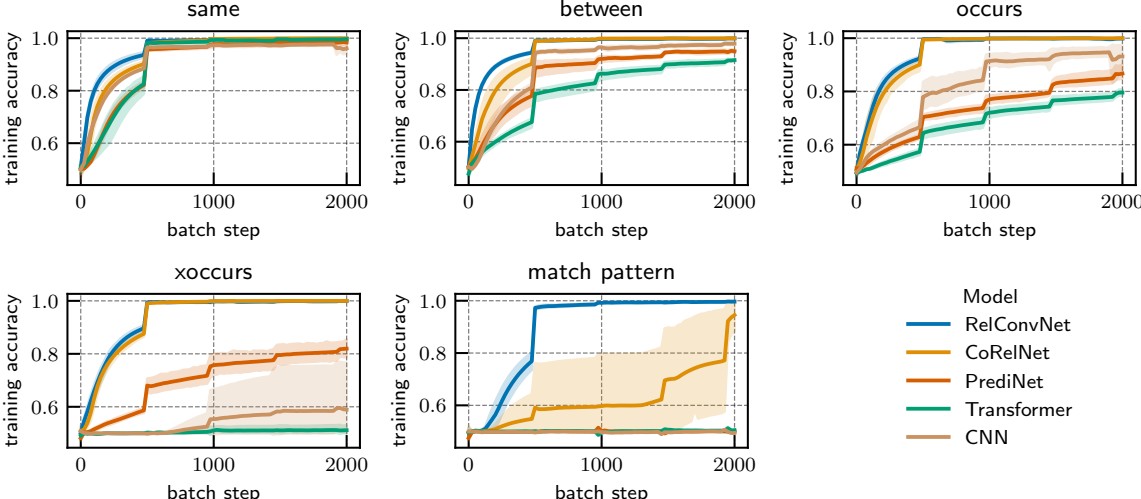

Figure 6: Training curves, up to 2,000 batch steps, for each relational games task. Solid lines indicate the mean over 5 trials and the shaded regions indicate a bootstrap 95% confidence interval. Note that this is in-distribution (i.e., on the "pentominoes" split).

**Learning groups via group attention.** Next, we analyze RelConvNet's ability to learn useful groupings through group attention in an end-to-end manner. We train a 2-layer relational convolutional network with 8 learned groups and a graphlet size of 3. We group based on position by using positional embeddings for $\texttt{key}(x_i)$. In Figure 7, we visualize the group attention scores $\alpha_{ij}^g$ (see Equation (6)) learned from one of the training runs. For each group $g \in [n_g]$, the figure depicts a $3 \times 3$ grid representing the objects attended to in that group. Since each group contains 3 objects, we represent the value $(\alpha_{ij}^g)_{i \in [3]}$ in the 3-channel HSV color representation.

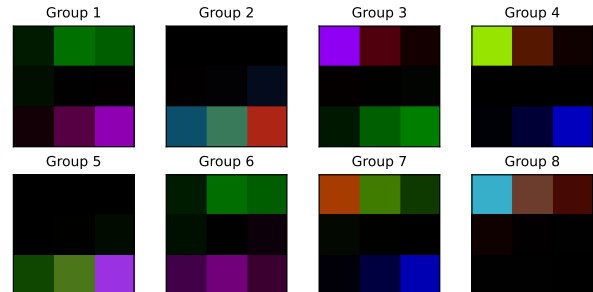

Figure 7: Learned groups in the 'match pattern' tasks by a 2-layer RelConvNet with group attention.

We observe that 1) group attention learns to ignore the middle row, which contains no relevant information; and 2) the selection of objects in the top row and the bottom row is structured. In particular, group 2 considers the relational pattern within the bottom row and group 8 considers the relational pattern in the top row, which is exactly how a human would tackle this problem. We refer to Figure 11 for an exploration of the effect of entropy regularization on group attention. We find that entropy regularization is necessary for the model to learn and causes the group attention scores to converge to interpretable discrete assignments.

### 4.2 *Set*: Grouping and Compositionality in Relational Reasoning

*Set* is a card game that forms a simple-to-describe but challenging relational task. The 'objects' are a set of cards with four attributes, each of which can take one of three possible values. 'Color' can be red, green, or purple; 'number' can be one, two, or three; 'shape' can be diamond, squiggle, or oval; and 'fill' can be solid, striped, or empty. A 'set' is a triplet of cards such that each attribute is either a) the same on all three cards, or b) different on all three cards.

In *Set*, the task is: given a hand of $n > 3$ cards, find a 'set' among them. Figure 8a depicts a positive and negative example for $n = 5$, with ∗ indicating the 'set' in the positive example. This task is deceptively challenging, and is representative of the type of relational reasoning that humans excel at but machine learning systems still struggle with. To solve the task, one must process the sensory information of individual cards to identify the values of each attribute, and then reason about the relational pattern in each triplet of

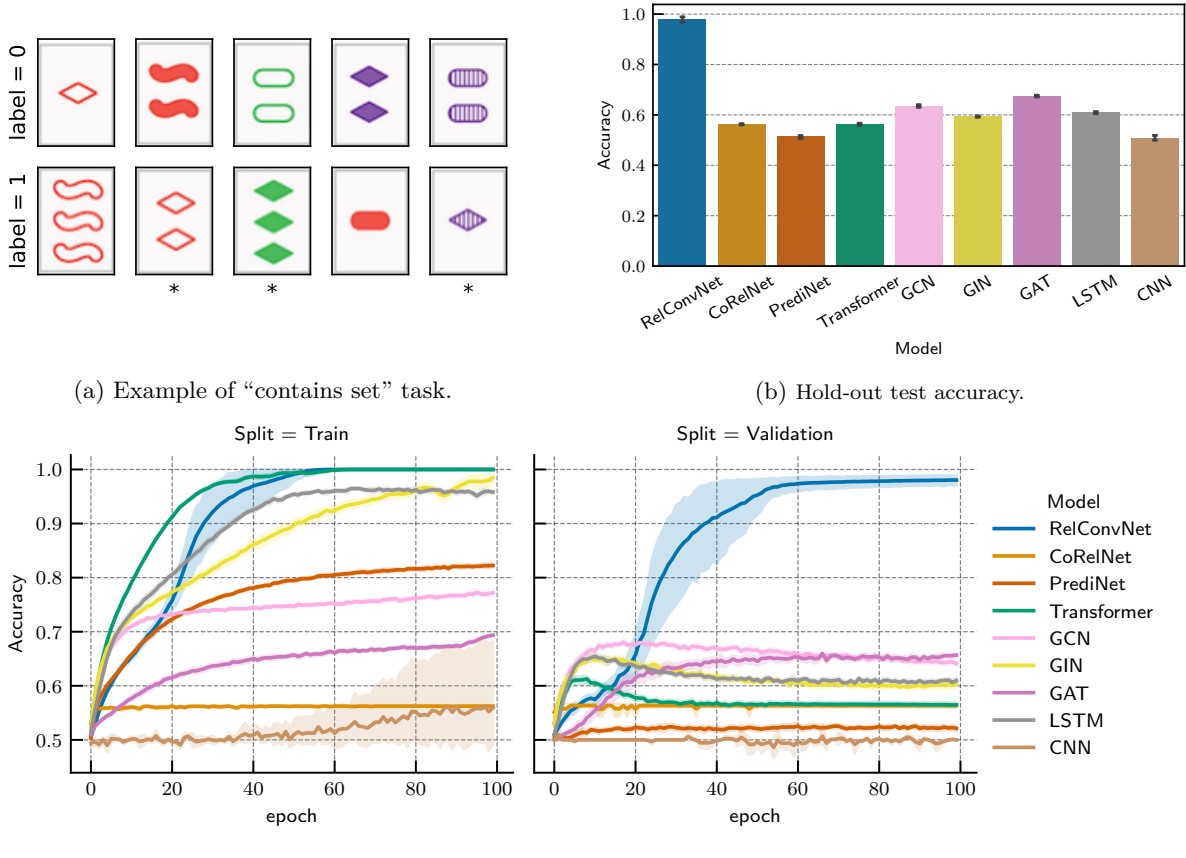

(a) Example of "contains set" task.

(b) Hold-out test accuracy.

(c) Training accuracy and validation accuracy over the course of training.

Figure 8: Results of "contains set" experiments. Bar height/solid lines indicate the mean over 10 trials and error bars/shaded regions indicate 95% bootstrap confidence intervals.

cards. The construct of relational convolutions proposed in this paper is a step towards developing machine learning systems that can perform this kind of relational reasoning.

In this section, we evaluate RelConvNet on a task based on *Set* and compare it to several baselines. The task is: given a collection of $n = 5$ images of *Set* cards, determine whether or not they contain a 'set'. All models share the common architecture $(x_1, \ldots, x_n) \to \texttt{CNN} \to \{\cdot\} \to \texttt{MLP} \to \hat{y}$, where $\{\cdot\}$ indicates the central module being tested. The CNN embedder is pre-trained on the task of classifying the four attributes of the cards and an intermediate layer is used to generate embeddings. The output MLP architecture is shared across all models. Further architectural details can be found in Appendix A.

In *Set*, there exists $\binom{81}{3} = 85\,320$ triplets of cards, of which $1\,080$ are a 'set'. We partition the 'sets' into training (70%), validation (15%), and test (15%) sets. The training, validation, and test datasets are generated by sampling $n$-tuples of cards such that with probability $1/2$ the $n$-tuple does not contain a set, and with probability $1/2$ it contains a set among the corresponding partition of sets. Partitioning the data in this way allows us to measure the models' ability to "learn the rule" and identify new unseen 'sets'. We train for 100 epochs with the same loss, optimizer, and batch size as the experiments in the previous section. For each model, we run 10 trials with different random seeds.

When using the default optimizer hyperparameters as in the previous experiment without hyperparameter tuning, we find that RelConvNet is the only model able to meaningfully learn the task in a manner that generalizes to unseen 'sets'. In particular, we observe that many baselines severely overfit to the training data, failing to learn the rule and generalize (see Appendix B.1). Although RelConvNet did not require hyperparameter tuning, we carry out an extensive hyperparameter sweep for all other baselines individually in order to validate our conclusions against the best-achievable performance for each baseline. We ran a total

of 1600 experimental runs searching over combinations of architectural hyperparameters (number of layers) and optimization hyperparameters (weight decay, learning rate schedule) individually for each baseline, with the goal of finding a hyperparameter configuration that is representative of the best achievable performance for each model class on this task. The results of the hyperparameter sweep are summarized in Appendix B.

Figure 8b shows the hold-out test accuracy for each model. Figure 8c shows the training and validation accuracy over the course of training. Here, RelConvNet uses the Adam optimizer with the default Tensorflow hyperparameters (constant learning rate of 0.001, $\beta_1 = 0.9, \beta_2 = 0.999$) while each baseline has its own individually-optimized hyperparameters, described in Appendix B.

We observe a sharp separation between RelConvNet and all other baselines. While RelConvNet is able to learn the task and generalize to new 'sets' with near-perfect accuracy (avg: 97.9%), no other model is able to reach a comparable generalization accuracy even after hyperparameter tuning. The next best is the GAT model (avg: 67.5%). Several models are able to fit the training data, reaching near-perfect training accuracy, but they are unable to "learn the rule" in a way that generalizes to the validation or test sets. This suggests that while these models are powerful function approximators, they lack the *inductive biases* to learn hierarchical relations.

In Figure 9 we analyze the geometry of the representations learned by the relational convolution layer. We consider all triplets of cards, compute the relation subtensor using the learned MD-IPR layer, and plot the relational inner product with the learned graphlet filter $\boldsymbol{f} \in \mathbb{R}^{s \times s \times d_r \times n_f}$. The result is a $n_f$-dimensional vector for each triplet of cards. We perform PCA to plot this in two dimensions, and color-code each triplet of cards according to whether or not it forms a 'set'. We find that the relational convolution layer learns a representation of the relational pattern in groups of objects that separates 'sets' and 'non-sets'. In particular, the two classes form clusters that are linearly separable even when projected down to two dimensions by PCA. This explains why RelConvNet is able to learn the task in a way that generalizes while the other models are not. In Appendix E we expand on this discussion,

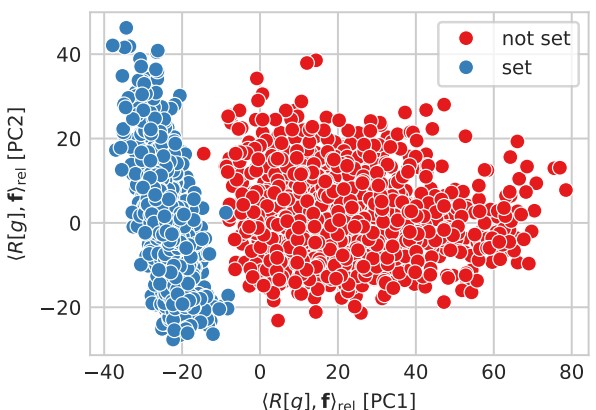

Figure 9: The relational convolution layer produces representations that separates 'sets' from 'non-sets'.

and further analyze the representations learned by the MD-IPR layer, showing that the learned relations map to the color, number, shape, and fill attributes.

It is perhaps surprising that models like GNNs and Transformers perform poorly on these relational tasks, given their apparent ability to process relations through neural message-passing and attention, respectively. We remark that GNNs operate in a different domain compared to relational models like RelConvNe, PrediNet, and CoRelNet. In GNNs, the relations are an input to the model, received in the form of a graph, and are used to dictate the flow of information in a neural message-passing operation. By contrast, in relational convolutional networks, the input is simply a set of objects without relations—the relations need to be *inferred* as part of the feature representation process. Thus, GNNs operate in domains where relational information is already present (e.g., analysis of social networks, biological networks, etc.), whereas our framework aims to solve tasks that rely on relations but those relations need to be inferred end-to-end. This offers a partial explanation for the inability of GNNs to learn this task—GNNs are good at processing network-style relations when they are given as input, but may not be able to infer and hierarchically process relations when they are not given. In the case of Transformers, relations are modeled implicitly to direct information retrieval in attention, but are not encoded explicitly in the final representations. By contrast, RelConvNet operates on collections of objects and possesses inductive biases for learning iteratively more complex relational representations, guided only by the supervisory signal of the downstream task.

Models like CoRelNet and PrediNet have relational inductive biases, but lack compositionality. On the other hand, deep models like Transformers and GNNs are compositional, but lack relational inductive biases. This experiment suggests that *compositionality and relational inductive biases are both necessary ingredients to efficiently learn representations of higher-order relations.* RelConvNet is a compositional architecture imbued with relational inductive biases and a demonstrated ability to tackle hierarchical relational tasks.

## 5 Discussion

**Summary**

In this paper, we proposed a compositional architecture and framework for learning hierarchical relational representations via a novel relational convolution operation. The relational convolution operation we propose here is a 'convolution' in the sense that it considers a patch of the relation tensor, given by a subset of objects, and compares the relations within it to a template graphlet filter via an appropriately-defined inner product. This is analogous to convolutional neural networks, where an image filter is compared against different patches of the input image. Moreover, we propose an attention-based mechanism for modeling useful groupings of objects in order to maintain scalability. By alternating inner product relation layers and relational convolution layers, we obtain an architecture that naturally models hierarchical relations.

**Discussion on relational inductive biases**

In our experiments, we observed that general-purpose sequence models like the Transformer struggle to learn tasks that involve relational reasoning in a data-efficient manner. The relational inductive biases of RelConvNet, CoRelNet, and PrediNet result in significantly improved performance on the relational games tasks. These models each implement different kinds of relational inductive biases, and are each designed with different motivations in mind. For example, PrediNet's architecture is loosely inspired by the structure of predicate logic, but can be understood as ultimately producing representations of pairwise difference-relations, with pairs of objects selected by an attention operation. CoRelNet is a minimal relational architecture that consists of computing an $n \times n$ inner product similarity matrix followed by a softmax normalization. RelConvNet, our proposed architecture, provides further flexibility across several dimensions. Like CoRelNet, it models relations as inner products of feature maps, but it achieves greater representational capacity by learning multi-dimensional relations through multiple learned feature maps or filters. More importantly, the relational convolutions operation enables learning higher-order relations between groups of objects. This is in contrast to both PrediNet and CoRelNet, which are limited to pairwise relations. Our experiments show that the inductive biases of RelConvNet result in improved performance in relational reasoning tasks. In particular, the *Set* task, where RelConvNet was the only model able to generalize non-trivially, demonstrates the necessity for explicit inductive biases that support learning hierarchical relations.

**Limitations and future work**

The tasks considered here are solvable by modeling only second-order relations. In the case of the relational games benchmark of Shanahan et al. (2020), we observe that the tasks are saturated by the relational convolutional networks architecture. While the "contains set" task demonstrates a sharp separation between relational convolutional networks and existing baselines, this task too only involves second-order relations. A more thorough evaluation of this architecture, and future architectures for modeling hierarchical relations, would require the development of new benchmark tasks and datasets that involve a larger number of objects and higher-order relations. This is a subtle and non-trivial task that we leave for future work.

The modules proposed in this paper assume object-centric representations as input. In particular, the tasks considered in our experiments have an explicit delineation between different objects. In more general settings, object information may need to be extracted from raw stimulus explicitly by the system (e.g., a natural image containing multiple objects in apriori unknown positions). Learning object-centric representations is an active area of research (Sabour et al., 2017; Greff et al., 2019; Locatello et al., 2020; Kipf et al., 2022), and is related but separate from learning relational representations. These methods produce a set of embedding vectors, each describing a different object in the scene, which can then be passed to the central processing module (e.g., a relational processing module such as RelConvNet). In future work, it will be important to explore how well RelConvNet integrates with methods for learning object-centric representations in an end-to-end system.

The experiments considered here are synthetic relational tasks designed for a controlled evaluation. In more realistic settings, we envision relational convolutional networks as modules embedded in a broader architecture. For example, a relational convolutional network can be embedded into an RL agent to enable performing tasks involving relational reasoning. Similarly, relational convolutions can perhaps be integrated into general-purpose sequence models, such as Transformers, to enable improved relational reasoning while retaining the generality of the architecture.

## Code and Reproducibility

The project repository can be found here: <https://github.com/Awni00/Relational-Convolutions>. It includes an implementation of the relational convolutional networks architecture, code and instructions for reproducing our experimental results, and links to experimental logs.

## Acknowledgment

This work is supported by the funds provided by the National Science Foundation and by DoD OUSD (R&E) under Cooperative Agreement PHY-2229929 (The NSF AI Institute for Artificial and Natural Intelligence).

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

## A   Experiments Supplement

### A.1   Relational Games (Section 4.1)

The pentominoes split is used for training, and the hexominoes and stripes splits are used to test out-of-distribution generalization after training. We hold out 1000 samples for validation (during training) and 5000 samples for testing (after training), and use the rest as the training set. We train for 50 epochs using the categorical cross-entropy loss and the Adam optimizer with learning rate 0.001, $\beta_1 = 0.9, \beta_2 = 0.999, \epsilon = 10^{-7}$. We use a batch size of 512. For each model and task, we run 5 trials with different random seeds.Table 1 contains text descriptions of each task in the relational games dataset in the experiments of Section 4.1. Table 2 contains a description of the architectures of each model (or shared component) in the experiments. Table 3 reports the accuracy on the hold-out object sets (i.e., the numbers depicted in Figure 5 of the main text). Figures 10 and 11 explore the effect of entropy regularization in group attention on learning using the "match pattern" task as an example.

| Task | Description |
|------|-------------|
| same | Two random cells out of nine are occupied by an object. They are the "same" if they have the same color, shape, and orientation (i.e., identical image) |
| occurs | The top row contains one object and the bottom row contains three objects. The "occurs" relationship holds if at least one of the objects in the bottom row is the same as the object in the top row. |
| xoccurs | Same as occurs, but the relationship holds if exactly one of the objects in the bottom row is the same as the object in the top row. |
| between | The grid is occupied by three objects in a line (horizontal or vertical). The "between" relationship holds if the outer objects are the same. |
| row match pattern | The first and third rows of the grid are occupied by three objects each. The "match pattern" relationship holds if the relation pattern in each row is the same (e.g., AAA, AAB, ABC, etc.) |

Table 1: Relational games tasks.

| Model / Component | Architecture |
|---|---|
| Common CNN Embedder | Conv2D → MaxPool2D → Conv2D → MaxPool2D → Flatten.
Conv2D: num filters = 16, filter size = $3 \times 3$, activation = relu.
MaxPool2D: stride = 2. |
| Common output MLP | Dense(64, 'relu') → Dense(2). |
| RelConvNet | CNN Embedder → MD-IPR → RelConv → Flatten → MLP.
MD-IPR: relation dim = 16, projection dim = 4, symmetric.
RelConv: num filters = 16, filter size = 3, discrete groups = combinations. |
| CoRelNet | CNN Embedder → CoRelNet → Flatten → MLP.
Standard CoRelNet has no hyperparameters. |
| PrediNet | CNN Embedder → PrediNet → Flatten → MLP.
PrediNet: key dim = 4, number of heads = 4, num relations = 16. |
| Transformer | CNN Embedder → TransformerEncoder → AveragePooling → MLP.
TransformerEncoder: num layers = 1, num heads = 8, feedforward intermediate size = 32, activation = relu. |
| GCN | CNN Embedder → AddPosEmb → (GCNConv → Dense) ×2 → AveragePooling → MLP.
GCConv: channels = 32, Dense: num neurons = 32, activation = relu |
| GAT | CNN Embedder → AddPosEmb → (GATConv → Dense) ×2 → AveragePooling → MLP.
GATonv: channels = 32, Dense: num neurons = 32, activation = relu |
| GCN | CNN Embedder → AddPosEmb → (GINConv → Dense) ×2 → AveragePooling → MLP.
GINConv: channels = 32, Dense: num neurons = 32, activation = relu |
| CNN | (Conv2D → MaxPool2D) ×8 → Flatten → Dense(128, 'relu') → Dense(2)
Conv2D: num filters = [16, 16, 32, 32, 64, 64, 128, 128], filter size = 3
MaxPool2D: stride = 2, apply every other layer. |

Table 2: Model architectures for relational games experiments.

| Task | Model | Hexos Accuracy | Stripes Accuracy |
|------|-------|----------------|------------------|
| same | RelConvNet | $0.989 \pm 0.002$ | $0.974 \pm 0.003$ |
|      | CoRelNet | $0.988 \pm 0.006$ | $0.724 \pm 0.112$ |
|      | PrediNet | $0.990 \pm 0.004$ | $0.983 \pm 0.007$ |
|      | Transformer | $0.997 \pm 0.001$ | $0.993 \pm 0.004$ |
|      | CNN | $0.990 \pm 0.001$ | $0.976 \pm 0.002$ |
| between | RelConvNet | $0.991 \pm 0.001$ | $0.988 \pm 0.002$ |
|      | CoRelNet | $0.995 \pm 0.001$ | $0.582 \pm 0.063$ |
|      | PrediNet | $0.978 \pm 0.006$ | $0.950 \pm 0.019$ |
|      | Transformer | $0.986 \pm 0.003$ | $0.961 \pm 0.010$ |
|      | CNN | $0.990 \pm 0.001$ | $0.971 \pm 0.003$ |
| occurs | RelConvNet | $0.980 \pm 0.001$ | $0.880 \pm 0.015$ |
|      | CoRelNet | $0.992 \pm 0.004$ | $0.518 \pm 0.012$ |
|      | PrediNet | $0.907 \pm 0.020$ | $0.775 \pm 0.046$ |
|      | Transformer | $0.881 \pm 0.015$ | $0.724 \pm 0.021$ |
|      | CNN | $0.984 \pm 0.008$ | $0.953 \pm 0.012$ |
| xoccurs | RelConvNet | $0.967 \pm 0.001$ | $0.946 \pm 0.006$ |
|      | CoRelNet | $0.980 \pm 0.007$ | $0.606 \pm 0.035$ |
|      | PrediNet | $0.872 \pm 0.036$ | $0.810 \pm 0.028$ |
|      | Transformer | $0.867 \pm 0.017$ | $0.753 \pm 0.031$ |
|      | CNN | $0.597 \pm 0.097$ | $0.590 \pm 0.084$ |
| match pattern | RelConvNet | $0.961 \pm 0.015$ | $0.870 \pm 0.041$ |
|      | CoRelNet | $0.942 \pm 0.011$ | $0.581 \pm 0.026$ |
|      | PrediNet | $0.710 \pm 0.040$ | $0.658 \pm 0.053$ |
|      | Transformer | $0.627 \pm 0.005$ | $0.591 \pm 0.006$ |
|      | CNN | $0.499 \pm 0.000$ | $0.489 \pm 0.000$ |

Table 3: Out-of-distribution generalization results on relational games. We report means $\pm$ standard error of mean over 5 trials. These are the numbers associated with Figure 5.

## A.2 *Set* (Section 4.2)

We train for 100 epochs using the cross-entropy loss. RelConvNet uses the Adam optimizer with learning rate 0.001, $\beta_1 = 0.9, \beta_2 = 0.999, \epsilon = 10^{-7}$. The baselines each use their own individually-tuned optimization hyperparameters, described in Appendix B. We use a batch size of 512. For each model and task, we run 5 trials with different random seeds. Table 4 contains a description of the architecture of each model in the "contains set" experiments of Section 4.2. Table 5 reports the generalization accuracies on the hold-out 'sets' (i.e., the numbers depicted in Figure 8b of the main text). Figure 12 explores the effect of different RelConvNet hyperparameters on the model's ability to learn the the *Set* task.

| Model / Component | Architecture |
|---|---|
| Common CNN Embedder | `Conv2D → MaxPool2D → Conv2D → MaxPool2D → Flatten → Dense(64, 'relu') → Dense(64, 'tanh')`.
`Conv2D`: num filters = 32, filter size = $5 \times 5$, activation = relu.
`MaxPool2D`: stride = 4. |
| Common output MLP | `Dense(64, 'relu') → Dense(32, 'relu') → Dense(2)`. |
| RelConvNet | `CNN Embedder → MD-IPR → RelConv → Flatten → MLP`.
`MD-IPR`: relation dim = 16, projection dim = 16, symmetric.
`RelConv`: num filters = 16, filter size = 3, discrete groups = combinations, symmetric relational inner product with 'max' aggregator. |
| CoRelNet | `CNN Embedder → CoRelNet → Flatten → MLP`.
Standard CoRelNet has no hyperparameters. |
| PrediNet | `CNN Embedder → PrediNet → Flatten → MLP`.
`PrediNet`: key dim = 4, number of heads = 4, num relations = 16. |
| Transformer | `CNN Embedder → TransformerEncoder → AveragePooling → MLP`.
`TransformerEncoder`: num layers = 2, num heads = 8, feedforward intermediate size = 128, activation = relu. |
| GCN | `CNN Embedder → (GCNConv → Dense) ×2 → AveragePooling → MLP`.
`GCConv`: channels = 128, `Dense`: num neurons = 128, activation = relu |
| GAT | `CNN Embedder → (GATConv → Dense) ×1 → AveragePooling → MLP`.
`GATonv`: channels = 128, `Dense`: num neurons = 128, activation = relu |
| GCN | `CNN Embedder → (GINConv → Dense) ×2 → AveragePooling → MLP`.
`GINConv`: channels = 128, `Dense`: num neurons = 128, activation = relu |
| CNN | `(Conv2D → MaxPool2D) ×10 → Flatten → Dense(128, 'relu') → Dense(2)`
`Conv2D`: num filters = [16, 16, 32, 32, 64, 64, 128, 128, 128, 128], filter size = 3
`MaxPool2D`: stride = [(2,2), NA, (2,2), NA, (2,2), NA, (2,2), (1,2), (1,2), (2, 2)] |

Table 4: Model architectures for "contains set" experiments.

| Model | Accuracy |
|---|---|
| RelConvNet | $0.979 \pm 0.006$ |
| CoRelNet | $0.563 \pm 0.001$ |
| PrediNet | $0.513 \pm 0.003$ |
| Transformer | $0.563 \pm 0.002$ |
| GCN | $0.635 \pm 0.003$ |
| GIN | $0.593 \pm 0.001$ |
| GAT | $0.675 \pm 0.002$ |
| LSTM | $0.609 \pm 0.002$ |
| CNN | $0.506 \pm 0.006$ |

Table 5: Hold-out test accuracy on "contains set" task. We report means $\pm$ standard error of mean over 10 trials. These are the numbers associated with Figure 8b.

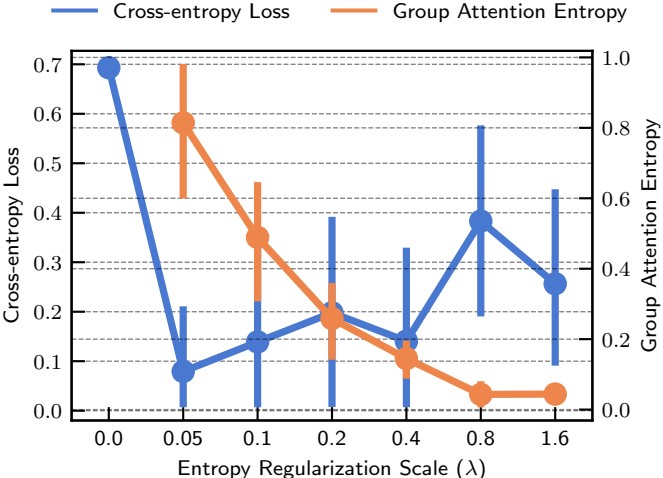

Figure 10: Trade-off between task loss and group attention entropy. RelConvNet models are trained on the "match pattern" task in the Relational Games benchmark varying the entropy regularization level. The overall model loss is $\mathcal{L}_{\texttt{loss}} + \lambda\mathcal{L}_{\texttt{entr}}$, where $\mathcal{L}_{\texttt{loss}} = \text{CrossEntropy}(y, \hat{y})$ is the task loss (blue line), $\mathcal{L}_{\texttt{entr}}$ is the entropy regularization term for the group attention scores (orange line) as defined in Section 3.2, and $\lambda$ is a scaling factor. Different lines correspond to different values of $\lambda$. When $\lambda = 0$ (no entropy regularization) the model fails to learn the task. A small amount of regularization is enough to guide the model to a good solution. Increasing $\lambda$ causes smaller group attention entropy at convergence, approaching discrete assignments.

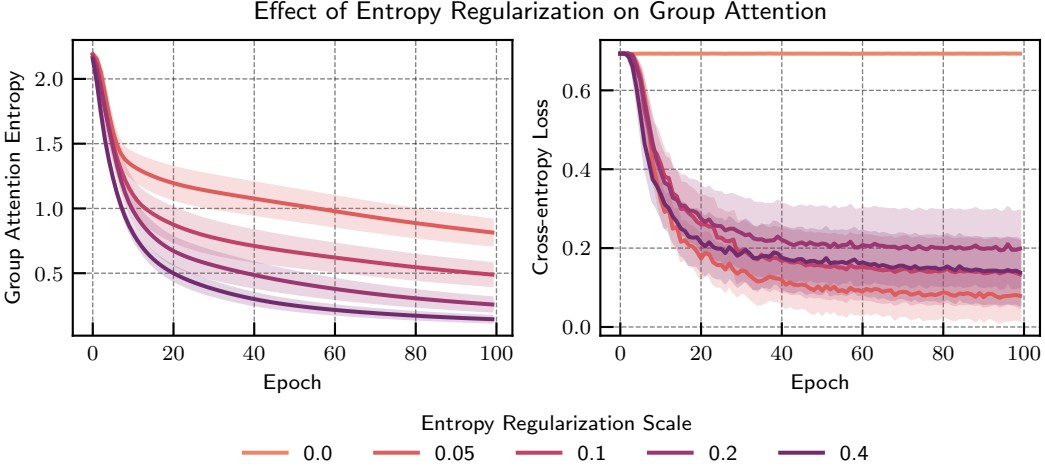

Figure 11: Effect of group attention entropy regularization. Group attention entropy (left) and baseline cross-entropy loss (right) of a relational convolutional network model trained on the "match pattern" task with different levels of entropy regularization. The overall model loss is $\mathcal{L}_{\texttt{loss}} + \lambda\mathcal{L}_{\texttt{entr}}$, where $\mathcal{L}_{\texttt{loss}}$ is the task loss, $\mathcal{L}_{\texttt{entr}}$ is the entropy regularization term, and $\lambda$ is a scaling factor. Different lines correspond to different values of $\lambda$. Without entropy regularization, the model fails to learn the task. With sufficient entropy regularization, the model is able to learn the task and group attention converges towards discrete assignments. The group attention entropy starts at $\log 9 \approx 2.2$ (the entropy of a uniform distribution) and decreases over the course of training. Expectedly, larger $\lambda$ values cause the entropy to decrease faster, converging towards a smaller value. When $\lambda$ is too large, the entropy regularization overwhelms the base cross-entropy loss and results in converging to a worse cross-entropy loss. Intuitively, one needs to strike a balance such that both the entropy regularization and the cross-entropy loss guide the evolution of the group attention map.

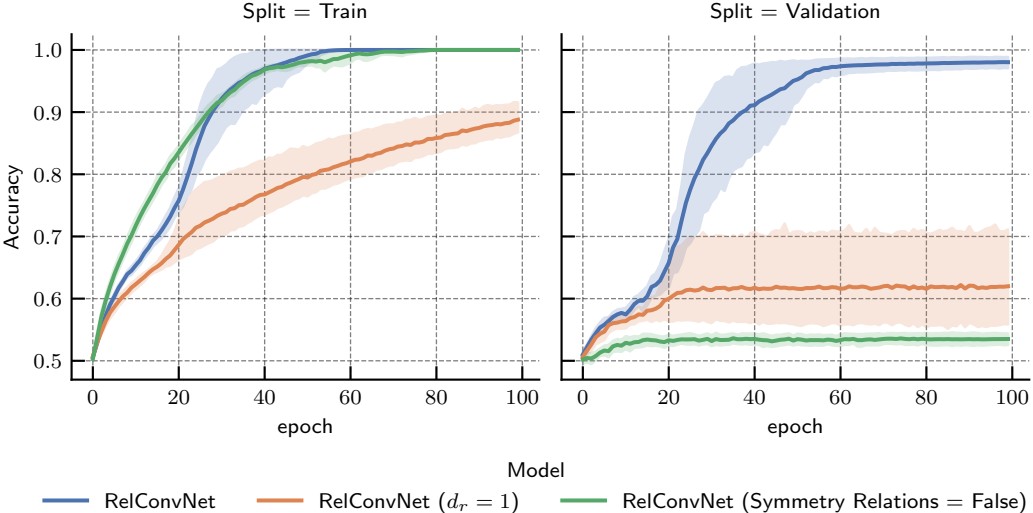

Figure 12: Exploring the effect of multi-dimensional relations and symmetric relations in RelConvNet. RelConvNet models matching the architecture described in Table 4 are trained on the *Set* task. We test two variants: 1) set the relation dimension to be $d_r = 1$ (instead of $d_r = 16$), and 2) remove the symmetry inductive bias (i.e., $W_1 \neq W_2$ in Equation (2)). We find that with $d_r = 1$ (which is analogous to CoRelNet's single-dimensional similarity matrix), the model struggles to find good solutions. In 10 different runs with random seeds, one run was able to find a good solution reaching an accuracy of 98.5%, whereas the other runs were stuck below 65%. This suggests that having multi-dimensional relations yields a more robust model with multiple different avenues for finding good solutions during the optimization process. In the case of the model with the asymmetric relations (i.e., lacking a symmetry inductive bias), the model is able to fit the training data, but fails to generalize. This suggests the symmetry is an important inductive bias for certain tasks.

## B  Hyperparameter sweep for baseline models

In order to ensure that we compare RelConvNet against the best-achievable performance by each baseline architecture, we carry out an extensive hyperparameter sweep over combinations of architectural hyperparameters and optimization hyperparameters. In particular, as seen in Appendix B.1, the baseline models severely overfit on the *Set* task, fitting the training data but failing to generalize to unseen 'sets'. Hence, we explore whether it is possible to avoid or alleviate overfitting through an appropriate choice of hyperparameters.

In Figure 13, we vary the number of layers in the baseline models to select an optimal configuration of each architecture. We find that increased depth beyond 2 layers is generally detrimental on this task. Based on these results, we choose the optimal number of layers as 2 for the Transformer, GCN, GIN baselines and 1 for the GAT baseline.

In Figure 14, we vary the level of weight decay. Expectedly, larger weight decay results in decreased training accuracy. Generally, weight decay has a small effect on validation performance (e.g., no discernable effect in CoRelNet or CNN). For some models, some choices of weight decay result in improved validation performance. Based on these results, we use a weight decay of 0 for CoRelNet/CNN, 0.032 for Transformer/GAT/GIN, and 1.024 for PrediNet/GCN/LSTM.

In Figure 15, we explore the effect of the learning rate schedule, comparing a cosine decay schedule against our default constant learning rate. For most models, there is no significant difference, with a constant learning rate sometimes slightly better. On the GAT model, however, the cosine learning rate schedule results in significantly improved performance. Based on these results, we use a cosine learning rate schedule for GAT and a constant learning rate for all other models.

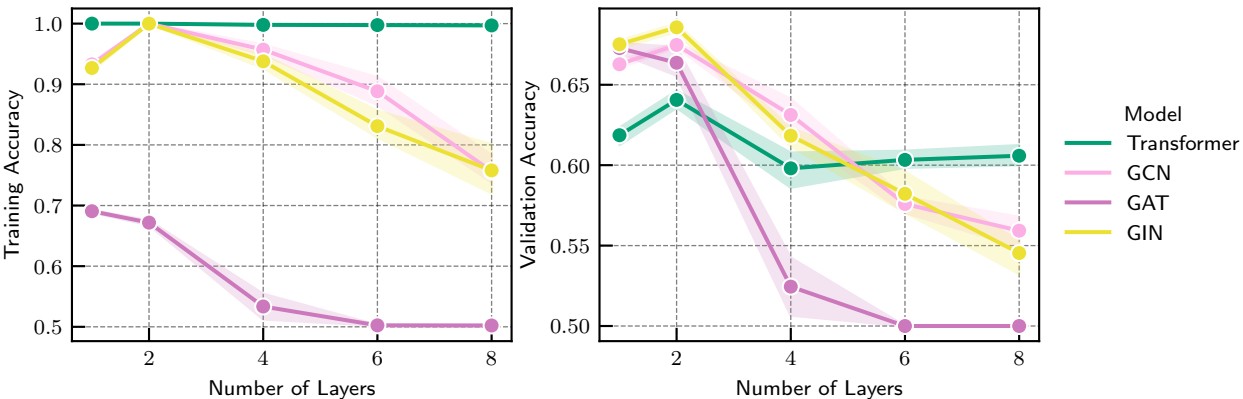

Figure 13: Hyperparameter sweep over number of layers in baseline architectures. Transformers and GNNs (e.g., GCN, GAT, GIN) are "compositional" deep learning architectures. Here, we explore the effect of depth (i.e., number of layers) on task performance for these baselines. The plots show the maximum training and validation accuracy reached throughout training for each depth (5 trials with different random seeds). Generally, we find that generalization performance drops with increasing depth. The optimal depth for the Transformer, GCN, and GIN models is 2 layers, and the optimal depth for the GAT model is 1-layer. The performance drop with depth can perhaps be attributed to increased difficulty of training and overfitting due to limited data and a lack of relational inductive biases. The AdamW optimizer is used with a constant learning rate of $10^{-3}$.

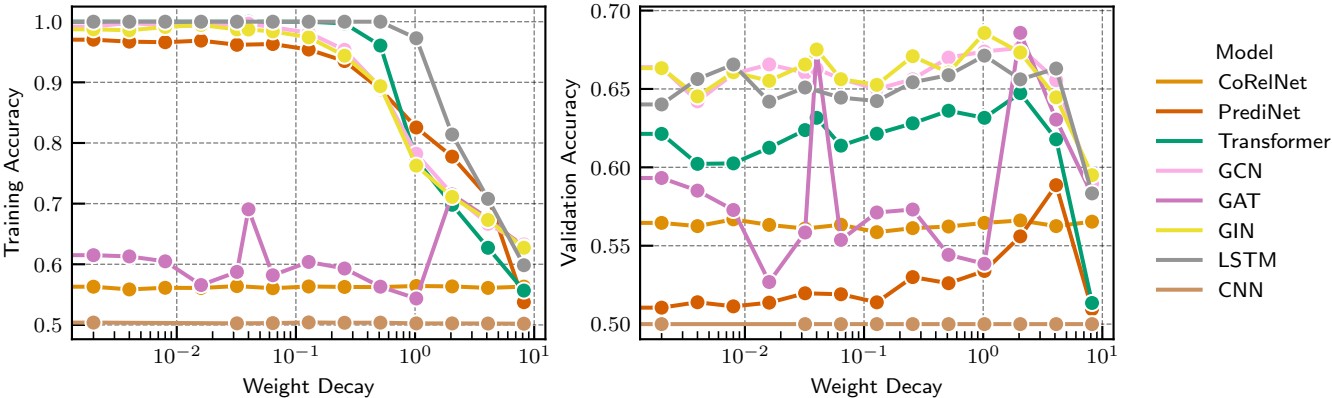

Figure 14: Hyperparameter sweep of weight decay in baseline architectures. To alleviate possible overfitting, we optimally tune a weight decay parameter for each model independently. We test weight decay values including 0 and $0.004 \cdot 2^i, i \in \{0, ..., 10\}$. We use the AdamW optimizer. The default weight decay in Tensorflow is 0.004.

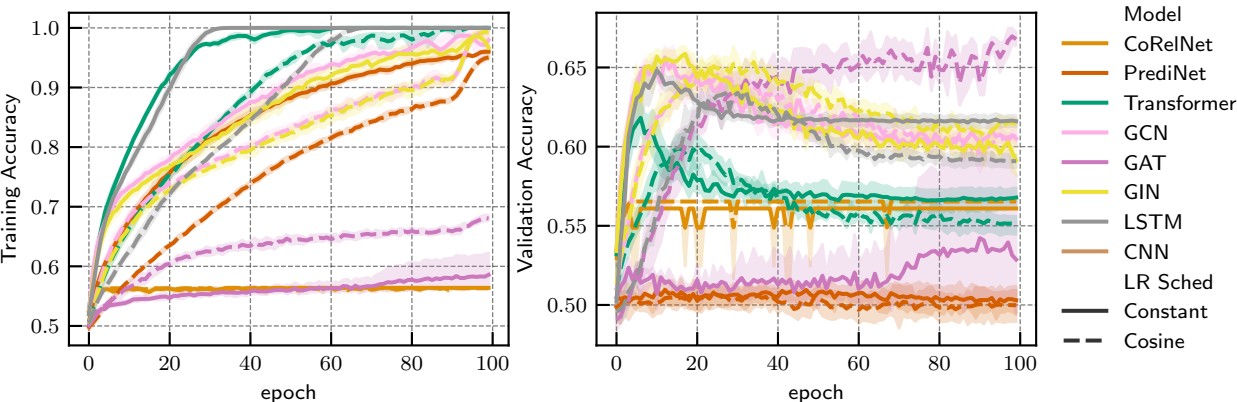

Figure 15: Hyperparameter sweep over learning rate schedule (constant vs cosine decay). We explore the effect of the learning rate schedule on model performance, comparing a constant learning rate against a cosine decay schedule. For most models, there is no significant difference, with a constant learning rate sometimes slightly better. On the GAT model, however, the cosine learning rate schedule results in significantly improved performance.

## B.1  Results without hyperparameter tuning

Figure 16 and Table 6 show the results of the *Set* experiment with a common default optimizer, without individual hyperparameter tuning.

|  | Accuracy |
| --- | --- |
| Model |  |
| RelConvNet | $0.979 \pm 0.006$ |
| CoRelNet | $0.563 \pm 0.001$ |
| PrediNet | $0.508 \pm 0.002$ |
| Transformer | $0.584 \pm 0.004$ |
| GCN | $0.595 \pm 0.003$ |
| GAT | $0.517 \pm 0.015$ |
| GIN | $0.590 \pm 0.003$ |
| LSTM | $0.602 \pm 0.003$ |
| GRU | $0.593 \pm 0.004$ |
| CNN | $0.506 \pm 0.006$ |

Table 6: Hold-out test accuracy on "contains set" task, with default optimizer hyperparameters. We report means $\pm$ standard error of mean over 10 trials. These are the numbers associated with Figure 16a.

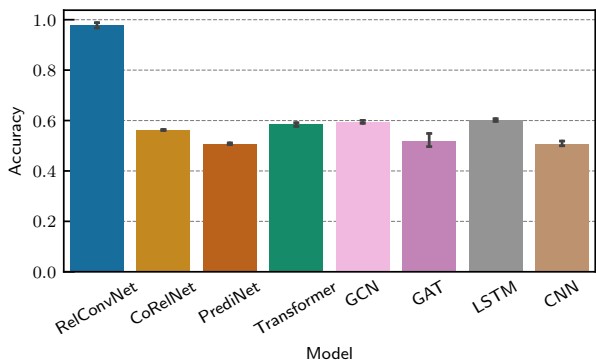

(a) Hold-out test accuracy.

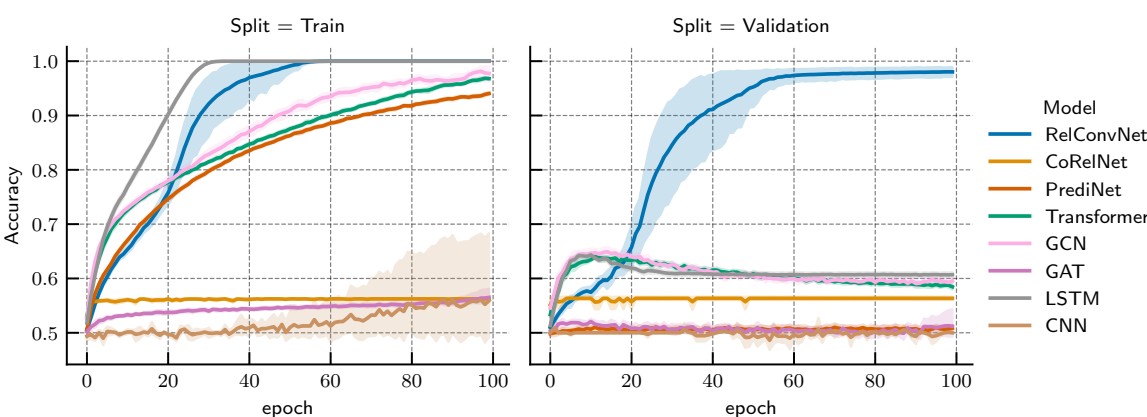

(b) Training accuracy and validation accuracy over the course of training, without hyperparameter tuning (i.e., Adam optimizer, no weight decay, constant learning rate of 0.001.

Figure 16: Results of "contains set" experiments with default optimization hyperparameters. Bar height/solid lines indicate the mean over 10 trials and error bars/shaded regions indicate 95% bootstrap confidence intervals.

## C   Discussion on use of TCN in evaluating relational architectures

In Section 4.1 the CoRelNet model of Kerg et al. (2022) was among the baselines we compared to. In that work, the authors also evaluate their model on the relational games benchmark. A difference between their experimental set up and ours is that they use a method called "context normalization" as a preprocessing step on the sequence of objects.

"Context normalization" was proposed by Webb et al. (2020). The proposal is simple: Given a sequence of objects, $(x_1, \ldots, x_m)$, and a set of context windows $\mathcal{W}_1, \ldots, \mathcal{W}_W \subset \{1, \ldots, m\}$ which partition the objects, each object is normalized along each dimension with respect to the other objects in its context. That is, $(z_1, \ldots, z_m) = \mathrm{CN}(x_1, \ldots, x_m)$ is computed as,

$$\mu_j^{(k)} = \frac{1}{|\mathcal{W}_k|} \sum_{t \in \mathcal{W}_k} (x_t)_j$$

$$\sigma_j^{(k)} = \sqrt{\frac{1}{|\mathcal{W}_k|} \sum_{t \in \mathcal{W}_k} \left( (x_t)_j - \mu_j^{(k)} \right)^2 + \varepsilon}$$

$$(z_t)_j = \gamma_j \left( \frac{(x_t)_j - \mu_j^{(k)}}{\sigma_j^{(k)}} \right) + \beta_j, \qquad \text{for } t \in \mathcal{W}_k$$

where $\gamma = (\gamma_1, \ldots, \gamma_d), \beta = (\beta_1, \ldots, \beta_d)$ are learnable gain and shift parameters for each dimension (initialized at 1 and 0, respectively, as with batch normalization). The context windows represent logical groupings of objects that are assumed to be known. For instance, (Webb et al., 2021; Kerg et al., 2022) consider a "relational match-to-sample" task where 3 pairs of objects are presented in sequence, and the task is to identify whether the relation in the first pair is the same as the relation in the second pair or the third pair. Here, the context windows would be the pairs of objects. In the relational games "match rows pattern" task, the context windows would be each row.

It is reported in (Webb et al., 2021; Kerg et al., 2022) that context normalization significantly accelerates learning and improves out-of-distribution generalization. Since (Webb et al., 2021; Kerg et al., 2022) use context normalization in their experiments, in this section we aim to explain our choice to exclude it. We argue that context normalization is a confounder and that an evaluation of relational architectures without such preprocessing is more informative.

To understand how context normalization works, consider first a context window of size 2, and let $\beta = 0, \gamma = 1$. Then, along each dimension, we have

$$\mathrm{CN}(x, x) = (0, 0),$$
$$\mathrm{CN}(x, y) = (\mathrm{sign}(x - y), \mathrm{sign}(y - x)).$$

In particular, what context normalization does when there are two objects is, along each dimension, output 0 if the value is the same, and $\pm 1$ if it is different (encoding whether it is larger or smaller). Hence, it makes the context-normalized output independent of the original feature representation. For tasks like relational games, where the key relation to model is same/different, this preprocessing is directly encoding this information in a "symbolic" way. In particular, for two objects $x_1, x_2$, context normalized to produce $z_1, z_2$, we have that $x_1 = x_2$ if and only if $\langle z_1, z_2 \rangle = 0$. This makes out-of-distribution generalization trivial, and does not properly test a relational architecture's ability to model the same/different relation.

Similarly, consider a context window of size 3. Then, along each dimension, we have,

$$\mathrm{CN}(x, x, x) = (0, 0, 0),$$
$$\mathrm{CN}(x, x, y) = \left( \frac{1}{\sqrt{2}} \mathrm{sign}(x - y), \frac{1}{\sqrt{2}} \mathrm{sign}(x - y), \frac{1}{\sqrt{2}} \mathrm{sign}(y - x) \right).$$

Again, context normalization symbolically encodes the relational pattern. For any triplet of objects, regardless of the values they take, context normalization produces identical output in the cases above. With context windows larger than 3, the behavior becomes more complex.

These properties of context normalization make it a confounder in the evaluation of relational architectures. In particular, for small context windows especially, context normalization symbolically encodes the relevant information. Experiments on relational architectures should evaluate the architectures' ability to *learn* those relations from data. Hence, we do not use context normalization in our experiments.

# D   Higher-order relational tasks

As noted in the discussion, the tasks considered in this paper are solvable by modeling second-order relations at most. One of the main innovations of the relational convolutions architecture over existing relational architectures is its compositionality and ability to model higher-order relations. An important direction of future research is to test the architecture's ability to model hierarchical relations of increasingly higher order. Constructing such benchmarks is a non-trivial task which requires careful thought and consideration. This was outside the scope of this paper, but we provide an initial discussion here which may be useful for constructing such benchmarks in future work.

**Propositional logic.** Consider evaluating boolean logic formula such as,

$$x_1 \wedge ((x_2 \vee x_3) \wedge ((\neg x_3 \wedge x_4) \vee (x_5 \wedge x_6 \wedge x_7))).$$

Evaluating this logical expression (in this form) requires iteratively grouping objects and computing the relations between them. For instance, we begin by computing the relation within $g_1 = (x_3, x_4)$ and the relation within $g_2 = (x_5, x_6, x_7)$, then we compute the relation between the groups $g_1$ and $g_2$, etc. For a task which involves logical reasoning of this hierarchical form, one might imagine the group attention in RelConvNet learning the relevant groups and the relational convolution operation computing the relations within each group. Taking inspiration from logical reasoning with such hierarchical structure may lead to interesting benchmarks of higher-order relational representation.

**Sequence modeling.** In sequence modeling (e.g., language modeling), modeling the relations between objects is usually essential. For example, syntactic and semantic relations between words are crucial to parsing language. Higher-order relations are also important, capturing syntactic and semantic relational features across different locations in the text and across multiple length-scales and layers of hierarchy (see for example some relevant work in linguistics Frank et al., 2012; Rosario et al., 2002). The attention matrix in Transformers can be thought of as implicitly representing relations between tokens. It is possible that composing Transformer layers also learns hierarchical relations. However, as shown in this work and previous work on relational representation, Transformers have limited efficiency in representing relations. Thus, incorporating relational convolutions into Transformer-based sequence models may yield meaningful improvements in the relational aspects of sequence modeling. One way to do this is by cross-attending to a the sequence of relational objects produced by relational convolutions, each of which summarizes the relations within a group of objects at some level of hierarchy.

**Set embedding.** The objective of set embedding is to map a collection of objects to a euclidean vector which represents the important features of the objects in the set (Zaheer et al., 2017). Depending on what the set embedding will be used for, it may need to represent a combination of object-level features and relational information, including perhaps relations of higher order. A set embedder which incorporates relational convolutions may be able to generate representations which summarize relations between objects at multiple layers of hierarchy.

**Visual scene understanding.** In a visual scene, there are typically several objects with spatial, visual, and semantic relations between them which are crucial for parsing the scene. The CLEVR benchmark on visual scene understanding (Johnson et al., 2017) was used in early work on relational representation (Santoro et al., 2017). In more complex situations, the objects in the scene may fall into natural groupings, and the spatial, visual, and semantic relations between those *groups* may be important for parsing a scene (e.g., objects forming larger components with functional dependence determined by the relations between them). Integrating relational convolutions into a visual scene understanding system may enable reasoning about such higher-order relations.

# E   Geometry of representations learned by MD-IPR and Relational Convolutions

In this section, we explore and visualize the representations learned by MD-IPR and RelConv layers. In particular, we will visualize the representations produced by the RelConvNet model trained on the *Set* task described in Section 4.2. Recall that the MD-IPR layer learns encoders $\phi_1, \psi_1, \ldots, \phi_{d_r}, \psi_{d_r}$. In this model $d_r = 16$, $\phi_i = \psi_i$ (so that learned relations are symmetric), and each $\phi_i$ is a linear transformation to $d_{\mathrm{proj}} = 4$-dimensional space. The representations learned by a selection of 6 encoders is visualized in Figure 17. For each of the 81 possible *Set* cards, we apply each encoder in the MD-IPR layer, reduce to 2-dimensions via PCA, and visualize how each encoder separates the 4 attributes: number, color, fill, and shape. Observe, for example, that "Encoder 0" disentangles color and shape, "Encoder 2" disentangles fill, and "Encoder 3" disentangles number.

Next, we visualize, we explore the geometry of learned representations of relation vectors. That is, the inner products producing the 16-dimensional relation vector for each pair of objects. For each $\binom{81}{2}$ pairs of *Set* cards, we compute the 16-dimensional relation vector learned by the MD-IPR layer, reduce to 2 dimensions via PCA, and visualize how the learned relation disentangles the latent same/different relations among the four attributes. This is shown in Figure 18. We see some separation of the underlying same/different relations among the four attributes, even with only two dimensions out of 16.

Finally, we visualize the representations learned by the relational convolution layer. Recall that this layer learns a set of graphlet filters $\boldsymbol{f} \in \mathbb{R}^{s \times s \times d_r \times n_f}$ which form templates of relational patterns against which groups of objects are compared. In our experiments, the filter size is $s = 3$ and the number of filters is $n_f = 16$. Hence, for each group $g$ of 3 *Set* cards, the relational convolution layer produces a 16-dimensional vector, $\langle R[g], \boldsymbol{f} \rangle_{\mathrm{rel}} \in \mathbb{R}^{n_f}$, summarizing the relational structure of the group. Of the $\binom{81}{3}$ possible triplets of *Set* cards, we create a balanced sample of "sets" and "non-sets". We then compute $\langle R[g], \boldsymbol{f} \rangle_{\mathrm{rel}}$ and reduce to 2 dimensions via PCA. Figure 9 strikingly shows that the representations learned by the relational convolution layer very clearly separate triplets of cards which form a set from those that don't form a set.

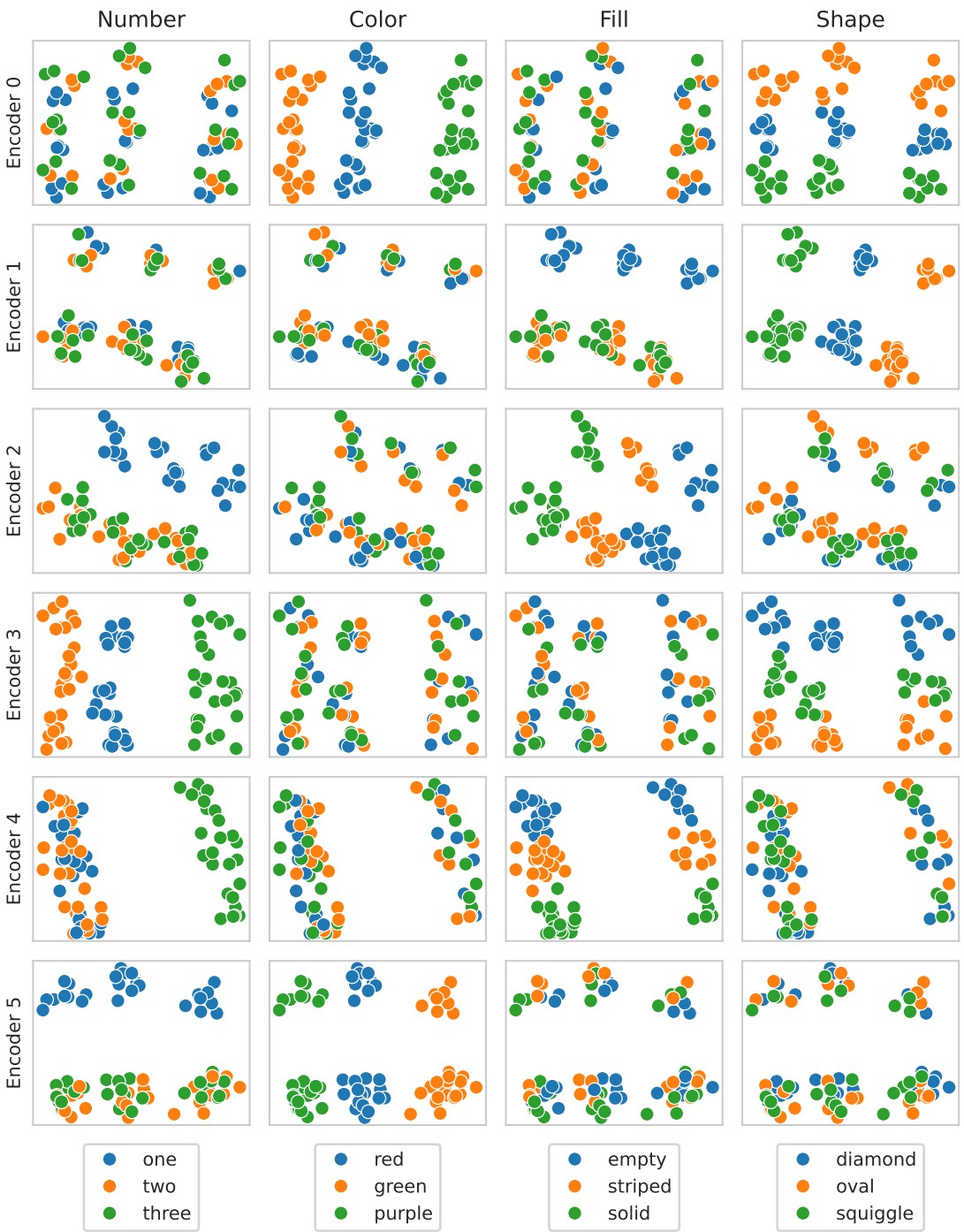

Figure 17: The encoders learned in the MD-IPR layer represent the latent attributes in the *Set* cards, with different encoders seemingly specializing to encode one or two attributes.

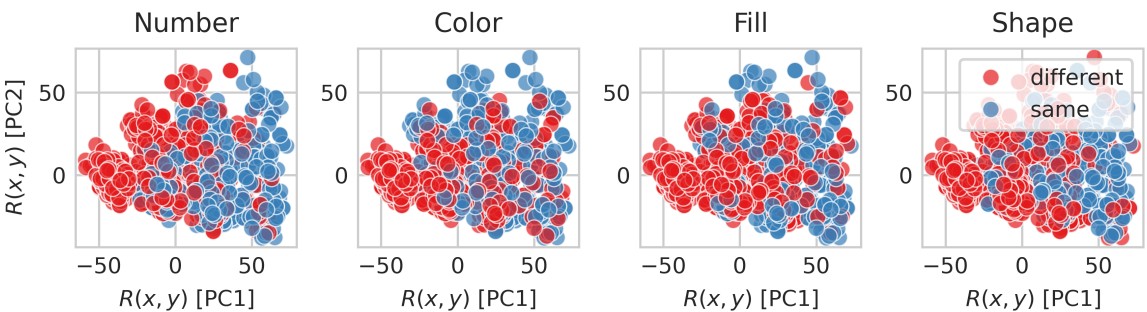

Figure 18: The relations learned by the MD-IPR layer encodes the latent relations underlying the *Set* task.

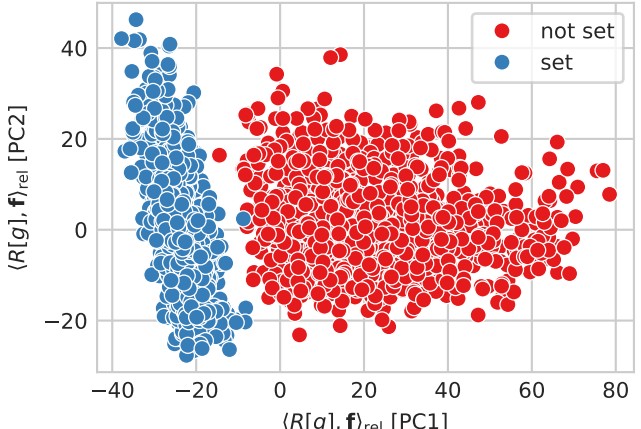

Figure 19: The relational convolution layer produces representations which separates 'sets' from 'non-sets'.

