# OpenReview forum: "Learning Hierarchical Relational Representations through Relational Convolutions"
_TMLR — Accepted by TMLR_

### Review · Reviewer_XLHi · 2024-07-03

**Summary Of Contributions:**

This paper introduces a novel type of DNN layer which performs _relational_ computations through inner products of embeddings. This is done by groups, and in a "convolutional" way, i.e. all groups are applied the same filters to. Fixed-sized groups are constructed via an attention mechanism which is regularized to have low entropy (i.e. picking objects rather than aggregating them).

The authors demonstrate good performance or relational tasks, as well as qualitatively different (and better) generalization properties on a set pattern matching task.

**Audience:**

Yes

**Broader Impact Concerns:**

Not addressed.

**Claims And Evidence:**

Yes

**Requested Changes:**

Things I'd like the authors to address:


$\left<x_1W_1,x_2W_2\right>=x_2W_2W_1^\top x_1^\top = x_2 W x_1$ has interesting properties when $W$ is positive definite, and also it's interesting that if $d_{proj} < d_\phi$ then $W_2W_1^\top$ is a low-rank approximation of $W$. I wonder if the authors have thought about this?

The text mentions that pairwise computations are a limitation, but really the number of possible groups is $2^n$ (or $\binom{n}{s}$ if we limit ourselves to size $s$ groups); isn't that the "scary number"? Transformers work "just fine" in $O(n^2)$ (and reasonable $O(n\log n)$ versions exist).

About the attention grouping, the text notes: "learn $n_g$ groupings of objects, retrieving $s$ objects per group." I'm a bit ambivalent w.r.t. describing this as "retrieving objects". In the limit, the attention operation in (6) can just average all the objects' embeddings, did it really "retrieve" something or did it do some intermediate aggregation operation? I don't think the way the authors wrote it is necessarily wrong but it feels misleading. In the proposed architecture, the relational convolutions are applied on aggregates, not directly on (groupings of) objects. The authors propose adding entropy regularization on the attention, but this entropy or the attention values are not reported (Figure 6 has no colorbar; no way to know exactly what those colors mean). How does entropy change during training? Are different factors more or less effective?

Regularizing attention entropy for sparsity is something that's been done before, please reference prior work.

One elephant in the room is what happens when objects are not neatly separated in the input space? or when the notion of what's an object or a collection of objects is blurry? Presumably this shouldn't be a problem, given the attention mechanism which is known to work even in e.g. vision tasks, but it would have been nice for the authors to train on even just a toy vision task (CIFAR100 SET?). As the authors point out in Limitations, this work is meant to lay the foundations, but this or experiments on >2 order interactions would add a lot to the work.

I wonder if a regular deep ConvNet baseline would make sense? One thing that the proposed architecture does is apply local filters to inputs. Sure the inputs of those tasks to a ConvNet would be biased by their location within the input, but that should be something that the model should be able to easily overcome with sufficient depth. The "Common CNN Embedder" described in Table 2 feels incredibly shallow.

The authors do a classic mistake, introduce a regularization term and not report results without the term applied/different weights on the term. More generally, many design choices appear to not be justified by empirical results (and I suspect they are, but the results just aren't reported in the paper. "Trust me it works" is not sufficient).

Final point, I know it may be unfair to compare to highly optimized transformer CUDA kernels, but I couldn't find it mentioned how the method fares computationally in practice. Does it seem to scale well?

**Strengths And Weaknesses:**

Generally the paper is well written, easy to understand, proposes an interesting and novel method, and provides strong basic evidence that the method performs well and does as intended.

Beyond the usual "there could be larger scale results", I think there could be more results demonstrating how the method works. The current investigative results very much pass the bar of TMLR to me, but I'd encourage the authors to dig even deeper (e.g. can the authors quantify and generalize what's in Figure 9? Compare it against other methods, including methods that do disentanglement?).

---

> ### Author Response · Authors · 2024-07-31
>
> Thank you for your review and your helpful thoughts and feedback.
>
> We have uploaded a revised version of the manuscript with several additions to address feedback from reviews. Please see the global response for an overview of these additions. Below, we will address your particular concerns and highlight additions related to your review in more detail.
>
> ---
>
> > $\langle x_1 W_1, x_2 W_2\rangle = x_2 W_2 W_1 x_1^\top = x_2 W x_1$ has interesting properties when $W$ is positive definite, and also it’s interesting that if $d_{\mathrm{proj}} < d_{\phi}$ then $W_2 W_1^\intercal$ is a low-rank approximation of $W$. I wonder if the authors have thought about this.
>
> Yes. Both of these points are important considerations. There are two parts to your question that we will respond to in turn. We refer to section 2 in the paper and provide a discussion below.
>
> ***Symmetric relations and positive (semi)-definiteness of $W$.***
> As you point out, when $W \in \mathbb{R}^{d \times d}$ is symmetric positive (semi-)definite, the relation $r(x_1, x_2) = x_1^\intercal W x_2$ has some additional structure. In the paper, we discuss this in terms of the *symmetry* of the relation $r(x, y)$. In our model, $W$ is parameterized by $W_1, W_2 \in \mathbb{R}^{d_{\mathrm{proj}} \times d}$ as $W:= W_1^{\intercal} W_2$. In the symmetric case, $W_1 = W_2$ and $W$ is symmetric positive semi-definite. This, of course, is a common way to parameterize PSD matrices (e.g., the Cholesky decomposition). Symmetry is an inductive bias that is useful in some tasks, but may be detrimental in others. In particular, as we mention in section 2, symmetric relations obey a transitivity property since $r(x, y)$ induces a pseudometric.
>
> We think of $W_1, W_2$ as feature filters that the inner product relation compares. When $W_1 = W_2$, the features extracted from $x_1$ is the same as the features extracted from $x_2$, and hence the relation $r(x_1, x_2)$ is comparing the same attributes across the two objects.  When $W_1 \neq W_2$, the relation $r(x_1, x_2)$ is asymmetric in the sense that it is comparing the two objects along different dimensions. For example, in the symmetric case, $r(x_1, x_2)$ may represent a relation like “the color is the same”, whereas in the asymmetric case, $r(x_1, x_2)$ may represent a relation like “$x_1$  is bigger than $x_2$”.
>
> Intuitively, symmetry is a useful inductive for the tasks we consider in our experiments since the task-relevant relations are symmetric similarity relations (e.g., in *Set*, the relevant relations are same/different across shape, color, number, and fill). The variant of RelConvNet in our experiments uses symmetric relations with $W_1 = W_2$. We have added further discussion to the paper on the effect of symmetry as an inductive bias for the tasks we consider. (More on this below.)
>
> ***Low-rank $W$.***
> The matrix $W$ will be at most rank $d_{\mathrm{proj}}$ when $d_{\mathrm{proj}} < d_{\phi}$ (which is always the case in our experiments). The rank of $W$ should be thought of as the dimensionality of the feature being filtered and compared. Typically, one would like to perform comparisons along particular attributes or “feature” (e.g., compare against color and shape separately). Recall that $W_1, W_2$ are projecting/extracting features to be compared from the original full feature representation. A “feature” is captured by a $d_{\mathrm{proj}}$-subspace of the original $d_\phi$-dimensional feature representation.
>
> We are unsure what you meant by $W_1^\intercal W_2$ being an “approximation” of something; perhaps that it is a parameterization of a low-rank matrix. But please let us know if there is a part of your question that we missed.

---

> ### Author Response · Authors · 2024-07-31
>
> > The text mentions that pairwise computations are a limitation, but really the number of possible groups is $2^n$ (or if we limit ourselves to size 𝑠 groups); isn't that the "scary number"? Transformers work "just fine" in $O(n^2)$ (and reasonable $O(n \log n)$ versions exist).
>
> Recall that in the case of discrete groupings (Section 3.1, Eq 5), the relational convolution takes the form
> $$R \ast \boldsymbol{f} := \left(\langle R[g], \boldsymbol{f} \rangle_{\mathrm{rel}}\right)_{g \in \mathcal{G}},$$
> where $\mathcal{G}$ is the set of discrete groups. The computational cost of this operation is $|\mathcal{G}| \cdot s^2 \cdot d_r \cdot n_f$, where $s$ is the graphlet filter size, $d_r$ is the relation dimension, and $n_f$ is the number of filters. If $\mathcal{G}$ is, say, all combinations of size $s$, then $|\mathcal{G}| = \binom{n}{s} \asymp n^s$. Typically, $s > 2$ (to be interesting, otherwise it is still representing a two-way relation), so this is worse than the $O(n^2)$ computational complexity of Transformers.
>
> *The bigger issue is that $|\mathcal{G}|$ becomes the number of objects at the next layer.* That is, the input to the next layer is a sequence of $|\mathcal{G}|$ objects, with the next layer then needing to compute a $|\mathcal{G}| \times |\mathcal{G}|$ relation tensor, etc. This causes a combinatorial explosion in the number of objects at deeper layers (e.g., number of objects at layer $\ell$ is $n_\ell = \binom{n_{\ell-1}}{s}$).
>
> Explicitly modeling the task-relevant groups as we propose enables us to keep the number of objects in each layer at a fixed amount, as a controllable hyperparameter.
>
> > About the attention grouping, the text notes: "learn $n_g$ groupings of objects, retrieving $s$ objects per group." I'm a bit ambivalent w.r.t. describing this as "retrieving objects". In the limit, the attention operation in (6) can just average all the objects' embeddings, did it really "retrieve" something or did it do some intermediate aggregation operation?
>
> Your point about terminology is well-taken. Of course, attention is not literally performing a discrete retrieval operation. It is rather a differentiable soft-retrieval operation. The softmax in attention is a differentiable version of an argmax, and in the limit (e.g., as temperature $1/\beta \to 0$) this recovers a hard-assignment. We use the term "retrieval" as an analogy to explain what we think of the attention step as implementing. We note also that the entropy regularization promotes learning an attention operation that is closer to a discrete assignment (see below).
>
> > How does entropy change during training? Are different factors more or less effective?
>
> We've added in the appendix an exploration of the effect of entropy regularization, at different levels of regularization. We focus our exploration on the "match pattern" task. We will summarize the results here. In Figure 9 of the updated manuscript, we explore the trade-offs between different levels of regularization on the task loss and group attention entropy at the end of training. We find that entropy regularization is necessary for learning the task. Without entropy regularization, the task loss does not decrease. But with even a small amount of regularization, the task loss is able to escape the local minimum at initialization. As the level of regularization increases, the group attention distribution's entropy at convergence decreases, converging to discrete assignments. In Figure 10, we plot the evolution of the task loss and entropy regularization loss over the course of training. The entropy regularization term starts at $\log(n)$ (the entropy of a uniform distribution) and decreases monotonically while the base cross-entropy loss also decreases monotonically. As expected, the entropy of the group attention scores at convergence is smaller when the regularization scaling factor is larger. At initialization, group attention is essentially computing an average with uniform weights, and the relations between these averaged objects contain too little information for the task loss to guide the model in a useful direction. However, with a little bit of entropy regularization, the model is able to escape the local minimum at initialization, at which point the task loss is able to guide the model toward a solution.
>
> > Regularizing attention entropy for sparsity is something that's been done before, please reference prior work.
>
> We've added some references.

---

> ### Author Response · Authors · 2024-07-31
>
> > One elephant in the room is what happens when objects are not neatly separated in the input space? or when the notion of what's an object or a collection of objects is blurry?
>
> This is an interesting and important question. Indeed, the modules we developed in this paper assume object-centric representations as input (e.g., a sequence of vector embeddings each corresponding to an object in the scene). The task of learning such object-centric representations from raw perceptual inputs is an active area of research. The problem of learning object-centric representations is related but separate to the problem of learning relational representations. In our paper, we tackle the latter.
>
> Learning object-centric representations can be done in an unsupervised way. For example, one notable piece of work in this area is Locatello et al.'s *Slot Attention*. The output of such methods is a sequence of vector embeddings, each describing an object in the scene. The output of something like Slot Attention can produce the input to relational convolutional networks to yield an end-to-end system learning from raw perceptual inputs. An important direction of future work will be to explore how well relational convolutional networks integrate with object-discovery methods like Slot Attention, and whether such a system could successfully learn to perform abstract relational reasoning in complex scenes.
>
> We have added a discussion on the connection between learning object-centric representations and learning relational representations [Section 5 of the updated manuscript].
>
> > I wonder if a regular deep ConvNet baseline would make sense? ... The "Common CNN Embedder" described in Table 2 feels incredibly shallow.
>
> First, a point of clarification. The "Common CNN Embedder" is a CNN module which is applied independently to each of the 9 objects in the $3 \times 3$ grid to produce a sequence of 9 embedding vectors. Each object (patch of image) is quite small; only $12 \times 12 \times 3$. So, a shallow CNN embedder is reasonable. The CNN only needs to extract very low-level features like color and shape.
>
> Nonetheless, your question stands as to whether a regular CNN could learn to perform these tasks end-to-end from the raw image input. We have carried out additional experiments to explore this. Interestingly, CNNs were not included as a baseline in the experiments of related work on relational architecture. There seems to be an implicit assumption in the literature on relational architectures that a CNN would be unable to learn such tasks since it lacks explicit relational processing capabilities. Given your question, we were curious to confirm whether this assumption was correct.
>
> We tested a 8-layer CNN model on the relational games tasks and a 10-layer model on the *Set* task, where the input to these models is now the raw image input. The results were interesting. On some of the relational games tasks (the simpler ones) the CNN model was perfectly capable of learning and generalizing. In fact, the CNN was on-par with the top-performing models on the "same" and "occurs" tasks. However, on the more difficult "xoccurs" and "match pattern" tasks, the CNN was the worst-performing model, with accuracy stuck at 50\%. Similarly, on the *Set* task, the CNN model was completely unable to learn the task in a manner that generalizes and, unlike certain other models like the Transformer, it was not even able to fit the training data.
>
> Overall, we observe in our experiments that certain simpler tasks are solvable by a broader range of architectures, giving little separation between different architectures, while more difficult tasks give a drastic separation (oftentimes, models completely fail to learn in a manner that generalizes). This highlights the need for additional relational reasoning benchmarks that enable further evaluation of relational architectures.
>
> The CNN baseline has been added to the experiments section. A discussion on these additional results has also been added to the paper.

---

> ### Author Response · Authors · 2024-07-31
>
> > Final point, I know it may be unfair to compare to highly optimized transformer CUDA kernels, but I couldn't find it mentioned how the method fares computationally in practice. Does it seem to scale well?
>
> At the end of Section 3.2, we discuss computational efficiency. We note that the overall computational complexity of a relational convolution layer (with group attention) is $O(n \cdot n_g \cdot s \cdot d + n_g \cdot s^2 \cdot d_r \cdot \max(d_{\mathrm{proj}}, n_f))$, where $n$ is the number of input objects, and the rest are hyperparameters. This can be implemented efficiently in modern deep learning libraries like PyTorch/Tensorflow/Jax/etc. In particular, computing the relation tensors can be computed in parallel with efficient matrix multiplications, and the group attention operation can use modern fast kernels like FlashAttention. In our experiments, the run-time of the RelConvNet experimental runs was similar to the Transformer baseline. Detailed experimental logs (including run-time, resource usage, metrics, etc.) are available through an online portal linked through the open-source code, which will be included in the de-anonymized version of the paper.

---

### Review · Reviewer_p6Dm · 2024-07-04

**Summary Of Contributions:**

The paper develops a method, termed relational convolutional networks, to learn representations of hierarchical relations. The authors propose an operation that convolve graphlet filters to incorporate the relational patterns in groups of objects. The final method enables compositional relational modules. Experiment results on various tasks or datasets are provided.

**Audience:**

Yes

**Claims And Evidence:**

Yes

**Requested Changes:**

- Since the method involves several modules, ablation studies on the contribution of each module would be beneficial.
- In the introduction, it is helpful to give a clear explanation of the originality of the work over existing ones (e.g., with respect to the problem setting considered or learning method).

**Strengths And Weaknesses:**

Strength:
- The paper is well written and clear.
- The authors provide a detailed explanation on the motivation of the model architecture.
- The proposed learning method is reasonable.
- The experiment results appear to support the effectiveness of the method.

Weakness:
- Only results on synthetic tasks are provided, though this may not be a huge issue given the lack of suitable benchmark.
- Ablation studies on each component are not provided (apologies if I missed it).
- The originality over existing works are not immediately clear from reading the introduction.

I am not sufficiently familiar with the literature and existing works to list other major weaknesses, and am open to hearing thoughts from other reviewers.

---

> ### Author Response · Authors · 2024-07-31
>
> Thank you for your review.
>
> We have uploaded a revised version of the manuscript with several additions to address feedback from reviews. Please see the global response for an overview of these additions. Below, we will address your particular concerns and highlight additions related to your review in more detail.
>
> > Since the method involves several modules, ablation studies on the contribution of each module would be beneficial.
>
> We have updated the paper to add some discussion of the effects of different configurations of the proposed module. In particular, the effect of the relation dimension $d_r$, imposed symmetry of the relations, entropy regularization, and others. For brevity, here we will highlight what we thought was most interesting. Please see Appendix A, Figures 9, 10, 11 for details.
>
> Recall that the *Set* task was by far the most difficult relational task in our experiments, with all other baselines failing to learn the task. Here, we discuss what factors contribute to the success of relational convolutional networks on the task.
>
> We find that the symmetry of relations is a crucial inductive bias. We experiment with variants of RelConvNet with symmetric ($W_1 = W_2$) and asymmetric ($W_1, W_2$ independent parameters) relations. We find that RelConvNet fails to learn the *SET!* task in a way that generalizes without the symmetry constraint---it is able to fit the training data but does not learn the rule in a way that generalizes.
>
> We also find multi-dimensional relations to be crucial for reliably learning this task. We compare a RelConvNet model with $d_r = 1$ to one with $d_r = 16$ (the variant reported in the paper). Interestingly, we find that in most trials, the $d_r = 1$ model gets stuck at around 50\% accuracy, but in a few trials, it does manage to learn the task reaching near-perfect accuracy. We have two intuitive explanations for this. The first is that this task of course relies on reasoning about relations across four different attributes, hence the underlying task-relevant relation is inherently multi-dimensional. The second is that having multiple relations (i.e., $d_r > 1$) gives the model multiple different avenues to find a solution, each initialized from a different starting point. When $d_r = 1$, the model may get stuck at a local minimum with no path to a good solution. But with multiple relations, a model is able to explore multiple avenues towards a solution making it much more likely to find a good one.
>
> Part of what makes this task very challenging is the need to perform some kind of combinatorial search with a limited supervision signal. This manifests itself in the shape of the accuracy curve over the course of training. It exhibits a "staircase" shape, suggesting that, when the model successfully learns the task, it does so all at once. We find that having symmetric and multi-dimensional relations is crucial for learning such tasks that require hierarchical combinatorial-like relational reasoning.
>
> These new results are depicted in Figure 11 in the appendix.

---

> ### Author Response · Authors · 2024-07-31
>
> > In the introduction, it is helpful to give a clear explanation of the originality of the work over existing ones (e.g., with respect to the problem setting considered or learning method).
>
> We will make a pass over the introduction to try to emphasize the originality of our work and its relation to existing work. Below is some discussion on this.
>
> Our work falls within a line of recent work concerned with developing neural network architectures with relational inductive biases for relational reasoning tasks. Representative examples of work on this problem include [1,2,3,4], and have influenced our thinking over the course of the project.
>
> In our work, the particular problem we tackle is developing neural architectures for learning *hierarchical* relational representations. Here, "hierarchical" means relations between relations. Previous work on relational architectures has been shallow and limited to first-order pairwise relations. Recognizing that deep learning's success stems from composing (simple) modules together to build iteratively more complex feature maps, we set out to develop a compositional relational architecture with the ability to learn hierarchical, higher-order relations.
>
> The way we propose doing this is through a novel operation that can be thought of as analogous to convolutions in a CNN---we call it *relational convolution*. Here, the key idea is to learn *graphlet filters* which represent templates of relational patterns between groups of objects and to "convolve" these patterns against the relations among objects in the input. This produces a sequence of embeddings, each representing the relational pattern in some grouping of objects. Repeating the operation produces higher-order representations of relations-between-relations.
>
> ---
>
> References:
>
> [1] Santoro, et al. "A simple neural network module for relational reasoning." NeurIPS 2017
>
> [2] Shanahan, et al. "An explicitly relational neural network architecture." ICML 2020
>
> [3] Webb, et al. "Emergent Symbols through Binding in External Memory." ICLR 2020
>
> [4] Kerg, et al. "On Neural Architecture Inductive Biases for Relational Tasks." ICLR 2022 Workshop OSC

---

### Review · Reviewer_gD6M · 2024-07-25

**Summary Of Contributions:**

This work investigates the impact of inductive biases on solving tasks relational tasks - whose problem instances are inputs containing multiple objects which may or may not be related in some way, and the task is to predict this relation (or its absence).

The work develops an architecture called Relational Convolutional Networks, which is designed to compositional in inferred relations, i.e. it has a compositionally and relational inductive bias. This proposed architecture is contrasted with architectures that are stated as only compositional (Transformers, GNNS) or only relational (PrediNet, CoRelNet), but not both.

The authors conduct experiments which support that both inductive biases are required for solving relational tasks.

**Audience:**

Yes

**Broader Impact Concerns:**

No broader impact concerns.

**Claims And Evidence:**

No

**Requested Changes:**

# Required (critical)

* Perform a reasonable hyperparameter sweep for the experimental section, critically including multilayer variants of all architectures, include the results of the hyperparameter sweep as an Appendix section. Optimization hyperparameters should be allowed to differ across methods. Optimization should be done using optimizers that use weight decay sensibly (e.g. AdamW), and a learning rate schedule should be used. The optimal weight decay may be zero, and other values should be investigated.
* Update main text results with best-in-class values for each method. An optional search constraint could be considering total-compute tied variants in order to be able to compare algorithms like-for-like.
* The hyperparameter search does not need to be exhaustive, but the results do need to support the primary claim of the paper (compositional and relational biases are both required for solving relational tasks) beyond reasonable doubt.


# Recommended for strengthening but not critical

* Remove instances of the word “novel” throughout.
* Consider renaming “Relational Convolutional Networks” as the naming is close to Relational Graph Convolutional networks which may cause confusion. Alternatively, highlight these are not the same early on.
* Clarify what is meant by hierarchical relation in the abstract - is it a hierarchy of element-wise relations or something else?
* Introduction paragraph 1: “...essential to accurately capturing complex systems.” Add citation.
* Introduction paragraph 1: “...building internal world models.” Add citation.
* Introduction paragraph 2: “...essential to success of deep representation learning.” Add citation.
* Introduction paragraph 2: “..flat first order architectures.” Add citation.
* Introduction paragraph 2: Add explanation here why Transformer is not a compositional framework for learning hierarchical relational representations. Add relevant citations.
* Bullet point end of page 1 “Grouping mechanisms”: Define “statistically intractable”.
* Page 2": "SET game". Add citation.
* Page 2: "Transformers". Add citation.
* Page 2: "Graph Neural Networks". Add citation.
* Section 2: Where discussing the proposal to model pairwise relations between objects, explain why Transformer and GNNs are insufficient, as both of these models naively sound like they would do this. Alternatively if the thinking is consistent with prior methods this should be stated here, and then specifically what is new in this work should be called out.
* Above Equation 2: Add elaboration why we want to promote weight sharing.
* Equation 3: the i, j indices at this part of the manuscript feel unusual since there should be a symmetry over i, j as the filter acts on elements of a set. This may cause the reader to expect the i, j insides on the relational filters to be redundant. The issue is re solved by the attention mechanism in Equation 6 which restores the permutation equivariance. It would be good to add here some commentary around the symmetry at this point, and that the x_i x_j being discussed at this point are abstract, and may be choosable in a way that restores the symmetry.
* Relate Equation 6 to the PerceiverIO Encoder with citations.
* Explain to what extent Named Entity Recognition is or is not a relational task, and how BERT’s solving Named Entity Recognition is consistent with your results.
* If after the hyperparameter sweep, GAT is still worse than GCN on training set, this should be explained.
* Add the number of layers used for every architecture to the hyperparameter tables 2 and 4.

**Strengths And Weaknesses:**

# Strengths

The work is well-written, easy to read, highly pedagogical. The figures and graphs are well-annotated and clear.

The work is also highly reproducible. with the majority of important hyperparameters exposed to the reader.

Experiments are accompanied by bootstrap CI estimates and multiple runs per point, lending trustworthiness to specific numbers quoted in the manuscript.

# Weaknesses

The main weakness of the paper lies in empirical evaluation.

The high-level problem is that broad generalizations about entire classes of architectures based on one hyperparameter set for each architecture, which are given for “Relational Games”  and “Set” in Tables 2 and 4 respectively. It is not stated how these hyperparameters were chosen, and critically (as is the primary interest for TMLR readers) whether these represent close to best-in-class choice for each problem setting.

A more specific problem for is that the RelConvNet architecture is explicitly designed as a two layer architecture in all experimental settings, first with the ML-IPR layer, followed by the RelConv layer. In contrast to the Transformer model being compared to, which has a single layer in each setting, as to the GNN variants. This lends a systematic advantage to the RelConvNet architecture being proposed, particularly when considering that the class of functions transformers represent on inputs changes significantly as the number of layers increases https://arxiv.org/abs/2209.11895. Additionally it is known that for 12 layer transformer models, relational NLP tasks like Named Entity Recognition are solved very well (https://arxiv.org/abs/1810.04805). This can be done in 6 layers with distillation procedures (https://arxiv.org/abs/1910.01108).

It is also unclear if standard optimization procedure is being followed, which would include a learning warmup period, and some form of decay schedule (cosine or liner) to a constant. No weight decay is used in the experimental settings, yet there is strong overfitting phenomena being observed in Figure 7. Similarly in Figure 7, we see GCN significantly outperforms GAT in terms of training accuracy - this should only happen if the optimization of the GAT model is being done poorly.

The remaining weakness (though less major) lies in Section 3, which builds up the proposed architecture. This section has no citations in it, which can give the reader an impression that this is all completely new, and can make it challenging to situate the proposed architecture amongst existing architectures and methods. For example, when talking about building pairwise relations, it could be useful to link to the transformer paper https://arxiv.org/abs/1706.03762, and when introducing Equation 6, it would be useful to relate this to a PerceiverEncoder https://arxiv.org/abs/2107.14795 where the learnable group query vector is the Perceiver Latent variable.

---

> ### Author Response · Authors · 2024-07-31
>
> Thank you for your review and constructive feedback.
>
> We have uploaded a revised version of the manuscript with several additions to address feedback from reviews. Please see the global response for an overview of these additions. Below, we will address your particular concerns and highlight additions related to your review in more detail.
>
> **Summary**
> - We carried out a hyperparameter sweep for each baseline model searching over combinations of architectural hyperparameters and optimization hyperparameters.
> - We updated the main text of the paper to compare the proposed RelConvNet architecture against optimized baselines. The RelConvNet model did not need hyperparameter tuning and continues to use the same default Adam optimizer.
> - The hyperparameter sweep resulted in marginally improved performance for the baselines, but does not change any conclusions. In particular, RelConvNet remains the only model able to learn the difficult *Set* task.
>
> **High-level concern about architecture and optimization hyperparameter choice**
> > The high-level problem is that broad generalizations about entire classes of architectures based on one hyperparameter set for each architecture
>
> The main high-level concern of this review is whether the conclusions drawn about the comparisons between our proposed architecture and the baseline architectures hold generally, or if they are specific to the specific architectural hyperparameters or optimization hyperparameters that we evaluate. We agree that it is important to validate whether the same conclusions continue to hold for different hyperparameter choices. In particular, the hyperparameter choice for each baseline should be approximately representative of what each model architecture is ideally capable of, for a given task.
>
> To further validate our conclusions, we perform an extensive hyperparameter sweep over optimization hyperparameters (Adam vs AdamW, weight decay, learning rate schedule) and architectural hyperparameters (number of layers) *individually for each model*. We note that this was not necessary for RelConvNet; we continue to use the Adam optimizer with a fixed learning rate, no weight decay, and the TensorFlow default hyperparameters. Our aim is to compare against the best-achievable performance for each baseline model class, giving the baselines the advantage of hyperparameter tuning which was unnecessary for RelConvNet. Appendix B in the updated manuscript describes the hyperparameter tuning process carried out for each baseline model architecture and the different optimization hyperparameters chosen for each model. We re-ran each baseline with the optimal hyperparameters and updated the results in the main text of the paper.
>
> The hyperparameter sweep resulted in marginally improved performance in the baselines, but did not change the general message of the experimental results. On the *Set* task, the best-performing baseline model (excluding RelConvNet) before hyperparameter tuning was the LSTM achieving a generalization accuracy of 60.2%; the best-performing model after the hyperparameter sweep is a 1-layer GAT achieving a generalization accuracy of 67.5\%. For most models, the improvement from hyperparameter tuning was small. For the GAT baseline, however, the difference was significant, rising from 51.7% to 67.5%. RelConvNet remains by far the best-performing model, achieving a generalization accuracy of 97.9\% (without tuning optimization hyperparameters).
>
> Below, we briefly summarize the results of the hyperparameter sweep with respect to the factors that you asked about.

---

> ### Author Response · Authors · 2024-07-31
>
> **Hyperparameter sweep procedure for Baselines.**
>
> We ran a total of 1620 experimental runs performing a hyperparameter sweep searching over combinations of architectural and optimization hyperparameters for all baselines with the goal of finding a hyperparameter configuration that is representative of the best achievable performance for each baseline. The results of this sweep are summarized in Appendix B. The full results and experimental logs of all runs will be made publicly available (through a W&B portal) in the de-anonymized final paper, same as the other experimental results from the paper.
>
>
> **The effect of depth on performance in baselines.**
> > In contrast to the Transformer model being compared to, which has a single layer in each setting, as to the GNN variants
>
> > Perform a reasonable hyperparameter sweep for the experimental section, critically including multilayer variants of all architectures, include the results of the hyperparameter sweep as an Appendix section.
>
> We vary the number of layers between 1 and 8 for each baseline. We find that increasing depth beyond two layers is generally detrimental. In the Transformer, GCN, and GIN the two-layer model slightly outperformed a one-layer model; in the GAT the one-layer model slightly outperformed the two-layer model. We choose the depth in the final model for each architecture accordingly.
>
> **Optimization Hyperparameters**
>
> > It is also unclear if standard optimization procedure is being followed
>
> > Optimization should be done using optimizers that use weight decay sensibly (e.g. AdamW), and a learning rate schedule should be used. The optimal weight decay may be zero, and other values should be investigated.
>
> As explained in the paper, our initial experiments used the Adam optimizer with a learning rate of $0.001$, $\beta_1 = 0.9, \beta_2 = 0.999, \epsilon = 10^{-7}$. These are commonly used settings and are the Tensorflow defaults for Adam. In the first version of the paper, the same optimization hyperparameters were used for all models and were not tuned for any model, making the comparison fair. Nonetheless, as you point out, it is possible that tuning these optimization hyperparameters might result in meaningful improvements in the baseline architectures.
>
> We performed a hyperparameter sweep with the AdamW optimizer and used a weight decay ranging between 0.004 and 1.024. We also explored whether a learning rate schedule such as linear warmup + cosine decay improved results.
>
> The detailed results are in the updated manuscript. Weight decay resulted in marginal improvements for some models, but no discernable improvement in others. The cosine learning rate schedule resulted in a significant improvement in the GAT model, but no improvement in the other baselines. The optimal weight decay level and learning rate schedule are chosen for each model according to these results.
>
> **On relational inductive biases.**
>
> Our results show that our initial conclusions broadly hold and that the baseline models' inability to solve the more complex relational tasks (e.g., *Set*) is not due to a poor choice of hyperparameters, but rather is due to the lack of appropriate inductive biases for learning complex relational representations. In particular, the *Set* task requires reasoning about relational patterns within groups of objects, which RelConvNet is able to explicitly capture, whereas the other baselines cannot. Moreover, the combinatorial nature that *Set* exhibits as a relational reasoning task implies a hard limit on the number of training examples and requires the model to learn to generalize from limited data. Although deep models like Transformers or GAT are highly expressive and are able to fit a large class of datasets, they lack an inductive bias to discover this type of relational reasoning procedure.
>
> We highlight that this conclusion is broadly consistent with previous work tackling relational tasks (e.g., [1,2,3]) where it is found that Transformers under-perform on relational tasks, especially in data-scarce settings. (These works don't evaluate against GNNs, but the same discussion applies.) This can be understood through the perspective of inductive biases. Since there exist many choices of parameters that fit the training data, we need *inductive biases* that help the trained model choose a sensible solution among these possible solutions. In the case of relational tasks, appropriate relational inductive biases are needed, and in our work, we focus on *hierarchical* relations.
>
> ---
> References:
>
> [1] Shanahan, et al. "An explicitly relational neural network architecture." ICML 2020
>
> [2] Webb, et al. "Emergent Symbols through Binding in External Memory." ICLR 2020
>
> [3] Kerg, et al. "On Neural Architecture Inductive Biases for Relational Tasks." ICLR 2022 Workshop OSC

---

> ### Author Response · Authors · 2024-07-31
>
> **Briefly, some minor points**
>
> > Add the number of layers used for every architecture to the hyperparameter tables 2 and 4.
>
> The number of layers is described in these tables. For example, "$\cdots \to (\mathtt{GCNConv} \to \mathtt{Dense}) \times 2 \to \cdots$" means that the GAT model has 2 layers.
>
> > For example, when talking about building pairwise relations, it could be useful to link to the transformer paper https://arxiv.org/abs/1706.03762
>
> > Where discussing the proposal to model pairwise relations between objects, explain why Transformer and GNNs are insufficient, as both of these models naively sound like they would do this.
>
> We'd like to refer you to the following paragraphs in the paper:
> 1. "It is perhaps surprising that models like GNNs and Transformers perform poorly on these relational tasks, given their apparent ability to process relations through ..." in Section 4.2
> 2. The "Discussion on relational inductive biases" subsection in the discussion.
>
> There, we discuss how models like Transformers and GNNs differ from models like RelConvNet/PrediNet/CoRelNet/etc in the way they process relations. For example, in Transformers, the pairwise relations are used only as an intermediate step in an information retrieval operation rather than manifesting in the resultant representations. By contrast, relational architectures like ours produce a relation-centric representation, which enables more efficient learning and better generalization.
>
> > Equation 3: the i, j indices at this part of the manuscript feel unusual since there should be a symmetry over i, j as the filter acts on elements of a set.  ...
>
> Depending on the task, symmetry over permutations of groups may or may not be a desirable property. For example, in the *Set* experiments of section 4.2, permutation-invariance is intuitively a useful property since "setness" is a permutation-invariant relational group property. By contrast, in the "match pattern" task, the property of having an AAA vs ABA vs AAB vs ABB vs ABC relation among triplets of objects is *not* permutation-invariant, and hence you would not want that type of symmetry in your representation.
>
> In our architecture, the "relational inner products" as defined in Equation 3 are not permutation-invariant by default (i.e., different permutations of objects can have different relational representations computed by a relational convolution). In the subsection "Symmetric relational inner products" in Section 3.2, we discuss a symmetric variant of the relational inner product. This is used for the *Set* experiments, but not the relational games experiments.

---

### Author Response · Authors · 2024-07-31

Dear reviewers,

Thank you for your constructive feedback. We have made several additions to the paper in response to your comments, and have made every effort to address each of the criticisms and concerns that were raised.  We will summarize the changes at a high-level in this global response and delve into more detail in the individual responses.

1. We have added an exploration of the effect of entropy regularization in group attention on training dynamics and model performance. We confirm that entropy regularization is necessary for escaping local minima at initialization. Higher regularization leads to lower entropy and sparser group attention scores, but good performance is achievable even with a small amount of regularization. [Appendix A, Figures 9 & 10]

2. We have added an exploration and discussion of the effect of the RelConvNet architecture hyperparameters on the ability to learn the challenging *Set* task. We find that multi-dimensional relations and a symmetry inductive bias are crucial for learning the task in a manner that generalizes. [Appendix A, Figure 11].

3. We added a CNN baseline to all experiments to explore a reviewer's question about whether a deep CNN could learn relational tasks end-to-end from raw image input. We find that the CNN model succeeds at learning the easier relational games tasks, but completely fails to learn the more challenging relational games tasks and the *Set* task. [Updated Section 4, Figures 4,5,7, Appendix A]

4. In our initial experiments, we used the same optimization hyperparameters for all models, without an individual hyperparameter sweep. A reviewer was appropriately critical of this setup. Based on the reviewer's suggestion, we carried out an extensive hyperparameter sweep over architectural hyperparameters and optimization hyperparameters for each baseline model individually. This makes it possible to compare against the best-achievable performance for each baseline model class, giving the baselines the advantage of hyperparameter tuning, which was unnecessary for RelConvNet. We find that some baselines benefited marginally from hyperparameter tuning (some more than others), but the message of our experimental results remains the same. In particular, RelConvNet remains the only model that can solve the *Set* task. [Description of hyperparameter sweep in Appendix B of updated manuscript; Results updated in Section 4]

Detailed descriptions of each of these changes to the paper are provided in the responses to the individual reviewers, where we also respond to each of the more minor comments that the reviewers made. We thank you again for helping to improve the quality of our work.

---

### Decision · Action_Editor_QGwq · 2024-08-30

**Recommendation:** Accept with minor revision

**Comment:**

Given that the paper presents a novel framework for learning hierarchical relations, and provides a reasonable amount of empirical validation, it is safe to say that it makes a clear contribution that is technically sound. The reviewers were also unanimous in recommending it for acceptance.

**Audience:**

Yes, the TMLR audience would be interested in this work, as it presents a novel solution to a challenging ML problem.

**Claims And Evidence:**

This paper presents a framework for learning to identify hierarchical relationships (here, using visual data). The core of the approach is a relational convolution operation which processes pairwise relations between objects (expressed via graphlet filters). These operations can then be stacked, with the goal of achieving hierarchical relational representations. The authors claim that their relational convolutional network framework is an effective means of modelling relational tasks that have hierarchical structure.

To support these claims the authors present experiments using the relational games benchmark and the game Set. They compare their relational convolution networks to five other baseline models, some explicitly designed to handle relational data (GNNs, PrediNet, CoRelNet) and some more generic (transformers, CNNs). The authors show that their approach outperforms these other approaches in terms of basic training efficacy and out-of-distribution generalisation.

The reviewers agreed that the paper was well-written and provides a novel solution to the problem it is attempting to address. They raised a number of concerns, particularly around the thoroughness of the empirical investigations (e.g. whether proper exploration of hyperparameters had been done and determining the effect of entropy regularisation in group attention). The authors responded to these concerns with a thorough rebuttal and additional experiments. Following this, the reviewers were unanimous that the claims had been sufficiently substantiated and the paper was appropriate for publication in TMLR.